# LeanFlex-GKP: Advancing Hassle-Free Structured Pruning with Simple Flexible Group Count

## Abstract

Densely structured pruning methods — which generate pruned models in a fully dense format, allowing immediate compression benefits without additional demands — are evolving due to their practical significance. Traditional techniques in this domain mainly revolve around coarser granularities, such as filter pruning, thereby limiting performance due to restricted pruning freedom. Recent advancements in *Grouped Kernel Pruning (GKP)* have enabled the utilization of finer granularities while maintaining a densely structured format. We observe that existing GKP methods often introduce dynamic operations to different aspects of their procedures at the cost of adding complications and/or imposing limitations (e.g., requiring an expensive mixture of clustering schemes) or contain dynamic pruning rates and sizes among groups that result in a reliance on custom architecture support for its pruned models. In this work, we argue that the best practice to introduce these dynamic operations to GKP is to make `Conv2d(groups)` (a.k.a. group count) flexible under an integral optimization, leveraging its ideal alignment with the infrastructure support of *Grouped Convolution*. Pursuing such a direction, we present a one-shot, post-train, data-agnostic GKP method that is more performant, adaptive, and efficient than its predecessors while simultaneously being a lot more user-friendly, with little-to-no hyper-parameter tuning or handcrafting of criteria required.

## 1 Introduction

Despite having a proven track record revolving around computer vision tasks, modern convolutional neural networks (CNNs) face deployment challenges for growing model capacities. To address this issue of over-parameterization, *network pruning* — a field studying how to insightfully remove components from the original model without significant degradation to its properties and performance — has undergone constant development for being an intuitive way of potentially reducing the computation and memory footprint required to practically utilize a model (Blalock et al., 2020).

In this work, we advance the progress on *Grouped Kernel Pruning (GKP)* (Zhong et al., 2022), a recently developed structured pruning granularity with many deployment-friendly properties, by investigating a common design choice among existing GKP methods: **dynamic operations** — which is an act of applying different operations to the same task (e.g., clustering CNN filters with various combinations of dimensionality reduction and clustering techniques, as in TMI-GKP (Zhong et al., 2022)). We find that current GKP designs tend to include such operations in a sub-optimal manner, resulting in various complications and limitations. As a solution, we propose that **the best practice to implement dynamic operations to GKP is to make `Conv2d(groups)` (a.k.a. group count) flexible** under an integral optimization, leveraging its ideal alignment with the existing and future infrastructure support of *Grouped Convolution* (Krizhevsky et al., 2012). Our empirical evaluation showcases that by making these group counts flexible, we can afford to "lean down" on the rest of the typical GKP procedures, and therefore obtain a new one-shot, post-train, data-agnostic GKP method that is more performant, adaptive, and efficient than its predecessors while simultaneously being a lot more user-friendly with little-to-no hyper-parameter tuning or handcrafted criteria required. We can concisely summarize our contribution as *"advancing hassle-free structured pruning,"* as suggested in the title.

Given that our work develops upon specific observations made on existing adaptations of **grouped kernel pruning (Zhong et al., 2022), a recently proposed structured pruning granularity with**

**limited exposure, we hereby provide a rather extensive background on the procedure at the risk of being redundant**. Additionally, we refer readers to Zhong et al. (2022) and He & Xiao (2023) for more information regarding different structured pruning granularities.

## 1.1 TRADING PERFORMANCE FOR DEPLOYABILITY: THE PRACTICAL ADVANTAGE OF STRUCTURED PRUNING

Under the general realm of network pruning, two categories of techniques have been proposed, which are commonly known as *unstructured pruning* and *structured pruning* (Mao et al., 2017; Blalock et al., 2020; He & Xiao, 2023). While it can be faithfully concluded that these two categories have very different focuses and approaches, there is, unfortunately, no universally agreed distinction between what pruning methods constitute structured pruning and what do not.

Nonetheless, the general understanding follows a performance-deployability trade-off: an unstructured pruning method typically tends to enjoy a higher degree of pruning freedom — and thus better performance — but it is done so at the cost of leaving the pruned network to be sparse without a reduction in size and consequently require special libraries or hardware support to realize compression/acceleration benefits (Yang et al., 2018) (e.g., weight pruning (LeCun et al., 1989)). Conversely, a structured pruning method often removes model components in groups that follow the architecture design of the original network, potentially resulting in a smaller network. Specifically, the majority of structured pruning methods (e.g., filter pruning (Zhou et al., 2016; Li et al., 2017)) are capable of delivering pruned models that are reduced in dimension yet entirely dense (a.k.a. *densely structured*) and therefore provide immediate compression benefits without additional demand.

## 1.2 EXPLORING STRUCTURED PRUNING WITH FINER GRANULARITIES: GROUPED KERNEL PRUNING (GKP)

To narrow the performance gap between unstructured and structured pruning methods, many structured pruning works have been exploring finer pruning granularities, which are often regarded as *intra-channel pruning* due to the two most prevalent structured pruning approaches — channel pruning and filter pruning — and essentially drive their pruning operations upon the in and out channels of the original CNN model.

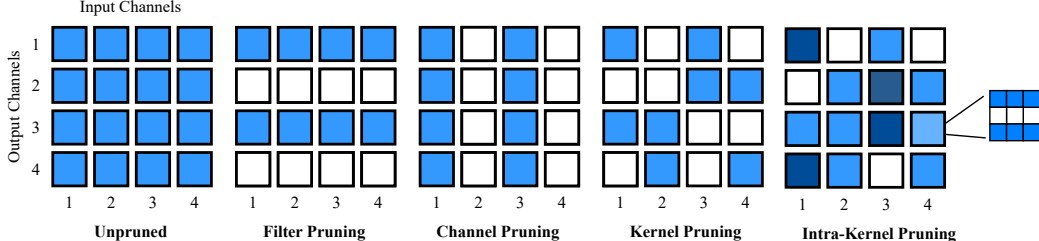

Figure 1: Different Structured Pruning Granularities

However, one major issue with these intra-channel explorations is that their pruned models are no longer dense and therefore lose the benefits of staying densely structured, such as increasing network efficiency without additional environment or hardware support (Yang et al., 2018). This is evident in Figure 1: it can be seen that if we naively seek out a finer pruning granularity than filter/channel pruning, we'd naturally have kernel pruning, which is intrinsically sparse. This is also the case for all *intra-kernel* pruning methods (e.g., stride pruning (Anwar et al., 2017), N:M sparsity (Zhou et al., 2021)), in which kernel-level sparsity is introduced. These methods might be "structured" by definition — they indeed "remove model components in groups" — but they often cannot provide efficiency benefits without external support due to the sparsity introduced to pruned models.

In order to achieve both increased pruning freedom and remaining densely structured, a special type of intra-channel pruning granularity called *Grouped Kernel Pruning (GKP)* (Zhong et al., 2022) has been proposed[1], where a finer pruning granularity than filter/channel pruning was achieved without introducing sparsity by leveraging the format of grouped convolution (Krizhevsky, 2014), as

---

[1]For the sake of rigor, this granularity was in fact revisited and refined by Zhong et al. (2022) at ICLR 2022 and coined as *grouped kernel pruning*. The granularity itself is, of course, naturally emerged in grouped convolution (Krizhevsky, 2014) and was first proposed under a pruning context by Yu et al. (2017); unfortunately, the proposal did not attract much traction (more about this in Appendix A).

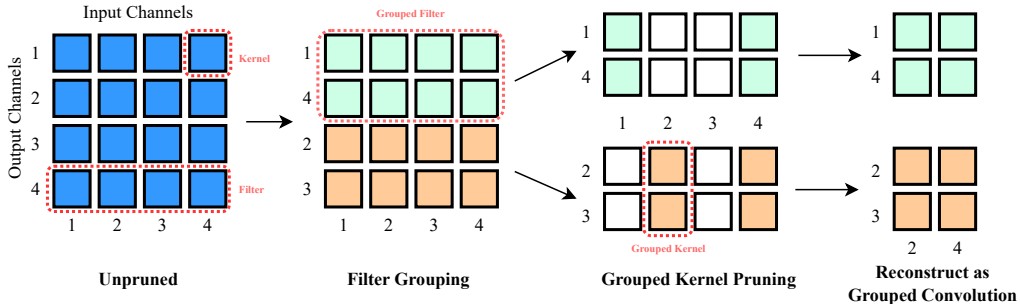

Figure 2: Procedure of Grouped Kernel Pruning

illustrated in Figure 2. To the best of our knowledge, GKP provides the highest degree of pruning freedom under the context of being densely structured and thus attracts the interests of the pruning community (Zhong et al., 2022; Zhang et al., 2022a; Park et al., 2023; He & Xiao, 2023).

### 1.3 A COMMON RECIPE FOR GKP-BASED METHODS: DYNAMIC OPERATIONS

Although GKP is still a fairly under-developed pruning granularity given its recency, we have observed a consistent pattern among the few existing successful works in GKP (e.g., TMI-GKP (Zhong et al., 2022) and DSP (Park et al., 2023)). Both methods introduce *dynamic operations* to different stages of its procedure and achieve significant performance improvement than GKP methods with only deterministic operations.

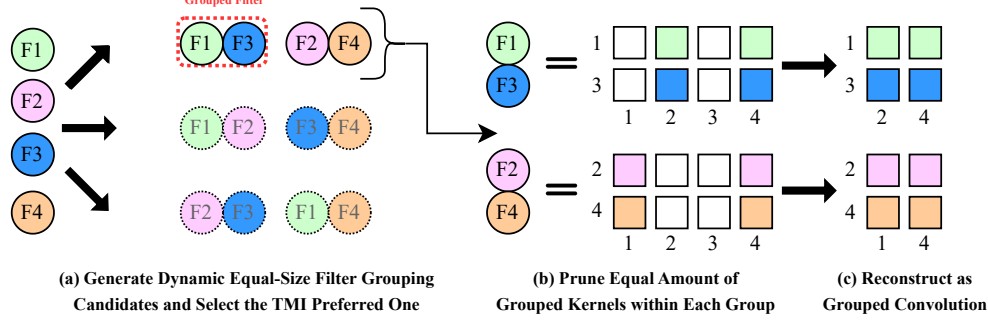

Figure 3: Procedure of TMI-GKP

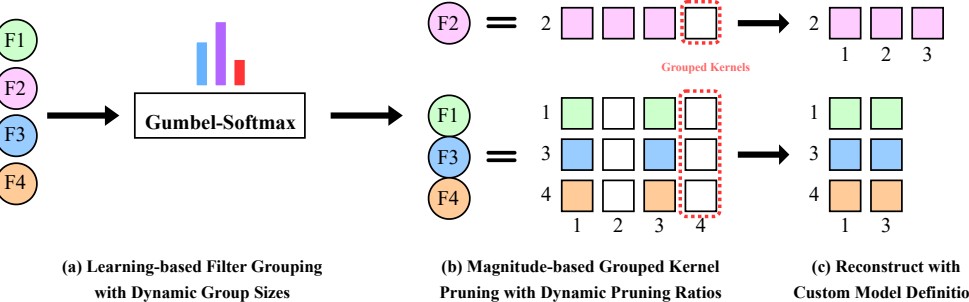

Figure 4: Procedure of Dynamic Structure Pruning (DSP)

As shown in Figure 3: TMI-GKP opts to include dynamic choices of clustering schemes in each of its convolutional layers. Similarly, in Figure 4, DSP makes its filter grouping and group kernel pruning stages dynamic in the sense that they may enjoy different group sizes and different in-group pruning rates for components within the same layer. While both methods deliver impressive performance, we notice that their adoption of dynamic operations results in various complications and limitations. For instance, several clustering schemes trialed in TMI-GKP can be very expensive to run. Yet, many of the produced clustering results are eventually discarded according to their tickets magnitude increase (TMI) scores. On the other hand, DSP essentially prunes grouped kernels in different sizes, where the resultant pruned network is irregularly shaped (i.e., having different dimensions of tensors within the same layer) and therefore relies on custom model definitions and convolutional operators to undergo training and inference — more on this in Section 2.1.

To mitigate the complications and limitations caused by dynamic operations in existing GKP methods, we propose a new method to include the dynamic operation within `Conv2d(groups)` (a.k.a. "group count" or "number of groups" as of grouped convolution). This means we allow each convolutional layer to take a flexible number of `groups` when grouping filters. **We argue this is the best area to integrate dynamic operations into a GKP procedure**, as this setup is directly supported by the well-adopted grouped convolution operator in modern ML frameworks, and is therefore able to make use of existing and future infrastructure updates and support for grouped convolutions. Empirical evaluation also supports the effectiveness of our approach.

Moreover, **after employing a flexible group count, we can simultaneously reduce the complexity and dependency of the rest of the GKP procedure and drastically improve the efficiency and usability of our method.** As an example, we utilize only one simple clustering operation rather than selecting one of the multiple expansive TMI-score-dependent clustering schemes. This makes our method usable without needing access to the training snapshots or checkpoints of the unpruned model (unlike TMI-GKP). This is a meaningful trait, given the prevalent utilization of pretrained models. We name our method LeanFlex-GKP, emphasizing that it is a GKP method that is more "leaned down" than others, utilizing flexible group counts as its primary mechanism.

We summarize the traits of our proposed method and the contributions of our work as follows:

- **Advancing the progress of GKP by identifying and solving a common pain point: dynamic operations**. We recognize the significance of dynamic operations to GKP, as well as the challenges of integrating them into a GKP procedure. By utilizing flexible group counts as a medium, we tactfully introduce such operations to our GKP procedure while avoiding the complications and limitations typically found in other GKP methods. Extensive empirical evaluation supports the effectiveness of our method.
- **Providing an efficient, hassle-free experience.** By reducing the complexity of various stages in the typical GKP procedure, our method provides a drastic advantage in terms of efficiency and adaptability. LeanFlex-GKP is a post-train, one-shot, data-agnostic procedure with little-to-no hyper-parameter tuning or setting handcrafting required, making it one of the most usable structured pruning methods.
- **Guiding future developments of GKP**. Aside from the proposed method itself, our work contains the most comprehensive empirical evaluation and ablation studies currently done on GKP. Given that GKP is an underdeveloped pruning granularity with many attractive properties, we believe our investigation may provide valuable insights and guidance to future scholars working to adopt GKP and its variants.

Due to our introduction's extensive coverage of tightly related pruning methods, we refer readers to Appendix A for more discussion on related works due to page limitation-related concerns.

## 2 MOTIVATION

### 2.1 FLEXIBLE GROUP COUNT AS THE DYNAMIC OPERATION IN GKP

As mentioned in Section 1.3, the involvement of dynamic operations plays a significant role to the GKP procedure. Yet, current GKP methods tend to adopt dynamic operations at the cost of adding complications or imposing limitations. Take TMI-GKP (Zhong et al., 2022) and Dynamic Structure Pruning (DSP) (Park et al., 2023) as examples: TMI-GKP trials different clustering schemes at its filter grouping stage — which consist of different combinations of various dimensionality reductions and clustering techniques — per each convolutional layer of the unpruned model, forming a dynamic choice of clustering schemes across the depth of the pruned model. DSP, with the term "dynamic" in its name, allows for dynamic group sizes and in-group pruning ratios upon the formed filter groups and thus enjoys a higher degree of pruning freedom than TMI-GKP.

While both methods demonstrate performance advantages over GKP methods with purely deterministic operations (e.g., KPGP by Zhang et al. (2022a) and many of the other alternative GKP procedures introduced in the appendix of Zhong et al. (2022)), the addition of such dynamic operations also comes with its own respective costs.

In TMI-GKP, some clustering schemes can be very expensive to run. e.g., $k$-PCA — one of the candidate dimensionality reduction techniques utilized in TMI-GKP — requires an eigen decomposition of

a convolutional layer's weight tensor, which is an expensive procedure requiring a complexity more than $\mathcal{O}(n^3)$ for a $n \times n$ matrix (Pan & Chen, 1999). Yet, all but a single produced clustering result are discarded if they are not preferred by its tickets magnitude increase (TMI) score: a weight-shift related metric inspired by series of works on the *lottery ticket hypothesis* (Frankle & Carbin, 2019). This makes the use of TMI-GKP challenging should the width of the target network become large.

In DSP, dynamic behavior is present in both the filter grouping and grouped kernel pruning stages, where the (learn-based) filter groups are allowed to be in different sizes, yet each filter group may opt to remove a different amount of grouped kernels, resulting in a pruning granularity that is finer than typical equal-group-equal-pruning-ratio GKP methods (Yu et al., 2017; Zhong et al., 2022; Zhang et al., 2022a). However, with the pruned network having different tensor shapes within the same layer, it can no longer be reconstructed into a grouped convolution format and instead relies on custom-defined model definitions and operators, therefore diminishing its practical adaptability.

In this work, we integrate dynamic operations on `Conv2D(groups)`; also commonly known as "group count" or "number of groups" under a grouped convolution context. This means we may group convolutions with different `groups` settings across model layers. We emphasize this setup is supported by the grouped convolution operator, and is therefore able to take advantage of the existing and future coming infrastructure updates and support. This flexible group count setup is different to that of TMI-GKP, where a hard-coded `groups=8` is applied for all models and layers, yet TMI-GKP decide the grouping result without consideration of the subsequent pruning (but our method does). Such a constant grouping schema and individual approach are revealed to be sub-optimal by our ablation studies in Appendix C. Our setup is also different from DSP, as the end results still have an equal group size and an identical pruning ratio among groups, and thus can be implemented without custom support.

## 2.2 Leaning Out for an Efficient GKP Procedure

Granted the effectiveness of flexible group counts, we may simultaneously afford to reduce the complexity of various GKP procedures. For example, instead of trialing different cluster schemes or employing a learn-based regularization procedure like TMI-GKP and DSP, we may simply utilize a $k$-Means$^{++}$ inspired clustering procedure to determine grouping, which drastically decreases the complexity and dependency requirement of filter grouping (Section 3.2).

During the grouped kernel pruning stage, methods like TMI-GKP formalize the procedure as a graph search problem and solve it with a multiple-restart greedy procedure, which is showcased to have a significant performance advantage over vanilla magnitudes or distance-based alternatives (Zhang et al., 2022a). However, we decided to use a tactfully designed distance and magnitude-based heuristic to achieve similar, if not better, accuracy retention rates to the unpruned models (Section 3.3). The removal of this procedure significantly reduces the runtime of our pruning procedure (as clocked in Table 7), and improves its usability on wide models.

## 2.3 Towards a Hassle-Free Experience

Although the after-prune performance and the efficiency of pruning procedures are certainly reasonable criteria when evaluating a method under a practical context, **usability across a broad scenario and being user-friendly along the process are another important set of factors to consider**. In fact, some of the most widely adopted pruning methods do not necessarily offer the best performance or the fastest runtime, but they are often extremely user-friendly as they can be run and deployed with minimal adjustments. Two examples of such work are OTOv2 (Chen et al., 2023) and DepGraph (Fang et al., 2023), which are architecture-agnostic methods capable of pruning any model, with OTOv2 capable of pruning from scratch.

Our method, LeanFlex-GKP, being a GKP method limited to CNNs, is not at the same level of generalization as OTOv2 or DepGraph. Still, we strive to maximize its usability under constraints by making it a post-train, one-shot, data-agnostic pruning method with standard fine-tuning procedures. This means as long as one has access to the weights of the CNN model and fine-tuning data is provided, one may adopt our pruning method to prune their model and fine-tune via standard SGD with no further interference. In comparison, previous GKP methods like TMI-GKP require access to the training snapshot/checkpoint of the unpruned model, and iterative GKP methods like DSP require regularization learning and pruning operations during the fine-tuning/retraining procedure.

On the note of user-friendliness, our method has little-to-no hyperparameters in place or handcrafted settings, making it extremely easy to use (and simultaneously reduces the human and resource effort of trial-and-error testing different settings). Furthermore, **the user of our method can reliably predict the pruned model size and computation requirement by simply multiplying the pruning rate by the original unpruned model**, making the whole pruning procedure a standardized and predictable experience. Note, this is a useful property surprisingly lacking in many modern pruning methods, such as Lin et al. (2020; 2019a); Park et al. (2023) and Chen et al. (2023), where the user will typically need to trial-and-error various hyperparameter combinations to achieve a certain pruning reduction. **We'd say the importance of being able to predictably obtain a pruned model at a certain size cannot be overly emphasized in a practical context**, as the alternative will require massive computation or even manual effort to search the suitable hyperparameter setting; sometimes, it is even impossible to prune to a specific reduction requirement.

## 3 PROPOSED METHOD

Our proposed method, LeanFlex-GKP, consists of a four-stage procedure:

1. **Filter grouping**: where we group filters within a certain convolution layer into $n$ equal-sized filter groups according to their distance towards $k$-Means$^{++}$ determined centers (Figure 5).

2. **Group kernel pruning:** where we prune a certain amount of grouped kernels out of all filter groups within the same layer. The pruning is determined by each grouped kernel's $L_2$ norm and distance to their geometric median (Figure 6).

3. **Post-prune group count evaluation:** where we evaluate all grouping and pruning strategies obtained under different group count settings and then select the one where the preserved group kernels have the maximum inter-group distance and the minimum intra-group distance (Figure 7).

4. **Grouped convolution reconstruction:** where we convert the pruned model to a grouped convolution format, just like we showcased in the standard GKP procedure (Figure 2).

As a general overview, the theme of our proposed method is to use the most lightweight and dependency-free measures to fulfill the purpose of each GKP stage. In the sections below, we will walk through the technicalities of our method, as well as demonstrate that **a SOTA-capable GKP method with many favorable properties can be forged by discerningly putting basic tools together and leveraging the power of flexible group counts**.

### 3.1 PRELIMINARIES

Suppose there is a convolutional neural network model $\mathbf{W}$ with $L$ convolutional layers, then the layer with index $l$ is denoted as $\mathbf{W}^l$. A layer can be viewed as a 4D tensor $\mathbf{W}^l \in \mathbb{R}^{C_{out}^l \times C_{in}^l \times H^l \times W^l}$, in which $C_{in}^l$ is the number input channels on layer $l$ (number of kernels in a filter), $C_{out}^l$ is the number output channels on layer $l$ (number of filters in a layer), and $H^l \times W^l$ is the kernel size. The task to perform a grouped convolution reconstruction upon $\mathbf{W}^l$, as illustrated in Figure 2, can be described as converting $\mathbf{W}^l$ to a $\mathbf{G}^l \in \mathbb{R}^{n \times C_{in}^l \times m \times H^l \times W^l}$, where $n$ stands for the group count setting of this conversion, and $m = C_{out}^l/n$ representing the group size.

### 3.2 KPP-AWARE FILTER GROUPING

The general goal of filter grouping is to cluster filters that are similar to each others within the same group, so that when such filters are "partially removed" due to pruning, the leftover components can hopefully cover the representation power of the removed components. In previous works like TMI-GKP (Zhong et al., 2022) and DSP (Park et al., 2023), this procedure is rather resource-intensive, with TMI-GKP trialing expensive clustering schemes under the guidance of its TMI score, and DSP employing a learning-based procedure.

To reduce the complexity of the grouping procedure, we designed a cost-effective $k$-Means$^{++}$ (KPP)-based filter clustering algorithm bolstered by two simple greedy strategies. Our procedure is illustrated in Figure 5. We denote $n$ to be the group count and $m = C_{out}^l/n$ to be the group size (number of filters within each filter group). In this particular visualization, we have $n = 3$ and $m = 4$. We demonstrate the efficiency and performance advantage of our method with wall-clock results in Table 7 and accuracy results in Table 2, support our claims made in Section 2.2 and Section 2.1.

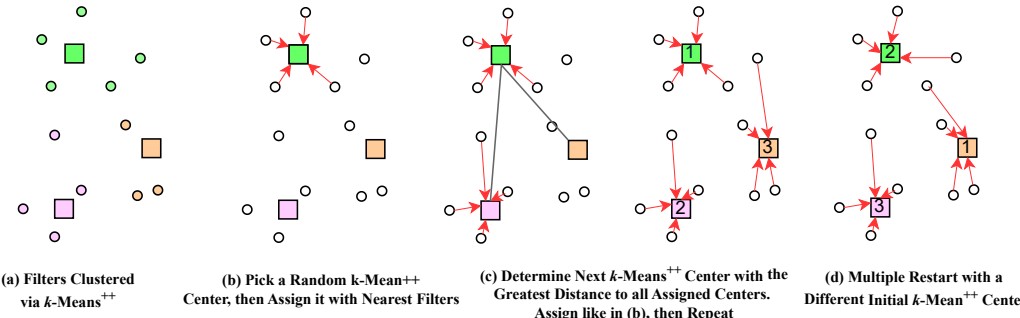

**(a) Filters Clustered via *k*-Means⁺⁺**  **(b) Pick a Random k-Mean⁺⁺ Center, then Assign it with Nearest Filters**  **(c) Determine Next *k*-Means⁺⁺ Center with the Greatest Distance to all Assigned Centers. Assign like in (b), then Repeat**  **(d) Multiple Restart with a Different Initial *k*-Mean⁺⁺ Center**

Figure 5: Visualization of the LeanFlex-GKP KPP-Aware Filter Grouping Procedure. We first cluster filters (the circles) via KPP into $n$ groups with no constraint on having an equal group size to determine clustering centers (the squares), as in (a). Then, our operation can be viewed as a cycle between assigning $m$ nearest filters into a KPP center to form a filter group, then finding the next KPP center to do subsequent filter assignments, as in (b) $\to$ (c); until $n$ filter groups are formed (the first KPP center is picked at random). Last, we conduct a multiple restart and repeat the (b)$\leftrightarrow$(c) center-finding-filter-assignments, as showcased in (d). After all multiple restarts, we are left with $n$ candidate filter grouping strategies, and select the strategy that has filters with the least intra-group distance to their respective KPP centers (having less summed length on red arrows).

### 3.3 $L_2$ & GEOMETRIC MEDIAN-BASED GROUPED KERNEL PRUNING

Previous methods like TMI-GKP converted its grouped kernel selection problem as a graph search problem, added with the help of a greedy procedure and multiple restarts. While such a procedure is generally efficient, it is still time and resource-consuming given a layer with a large amount of `in_channels`. Thus, inspired by the toolsets proposed in FPGM (He et al., 2019), we utilize a simple combination of $L_2$ norm and Geometric Median-based distance to form a lightning-fast pruning procedure, as illustrated in Figure 6. Again, we demonstrate the efficiency and performance advantage of our method with Table 7 and Table 3, support our claims made in Section 2.2 and Section 2.1.

Again, we demonstrate the efficiency advantage of our method with Table 7, as we claimed in Section 2.2.

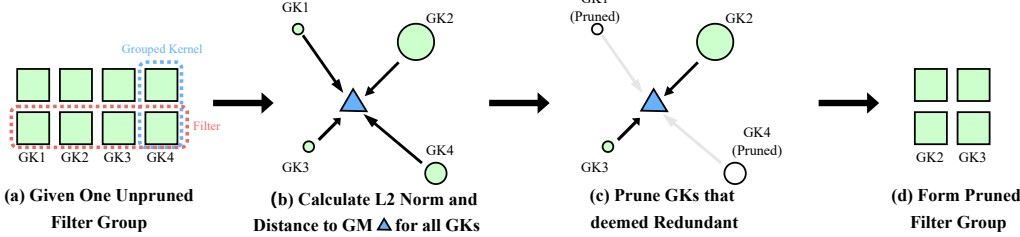

**(a) Given One Unpruned Filter Group**  **(b) Calculate L2 Norm and Distance to GM △ for all GKs**  **(c) Prune GKs that deemed Redundant**  **(d) Form Pruned Filter Group**

Figure 6: Visualization of LeanFlex-GKP $L_2$ & Geometric Median-based Grouped Kernel Pruning Procedure. Given an unpruned filter group as in (a), we first calculate the Geometric Median (GM) of its Grouped Kernels (GKs), as well as each GK's distance to the GM and their $L_2$ norm. These distances and the $L_2$ norm are visualized in (b) as the length of black arrows and the area of green circles, respectively. The GKs with large $L_2$ norms and small distances to their GMs are preserved and eventually reconstructed to the grouped convolution format, as shown (c) to (d) — please refer to Appendix B.1 for details.

### 3.4 POST-PRUNE GROUP COUNT EVALUATION

One primary motivation for our work is that our method makes use of flexible group counts under a GKP procedure. However, it is intrinsically challenging to evaluate clustering quality under different group counts (e.g., previously, Zhong et al. (2022) suggests metrics like a Silhouette score have little bearing under a network pruning context). Thus, we simply employ another Geometric Median-based evaluation as we have already done so in Section 3.3. We illustrate our evaluation as Figure 7 and provide **a walk-through of the complete LeanFlex-GKP procedure in pseudocode as Algorithm 1**. Given each group count evaluation is conducted upon a pruned convolutional layer (after being grouped with different `Conv2d(groups)`), our method makes connections between the (originally independent) filter grouping and grouped kernel pruning stage. Ablation study results in Table 4 confirm the advantage of this integral optimization design over other alternative setups.

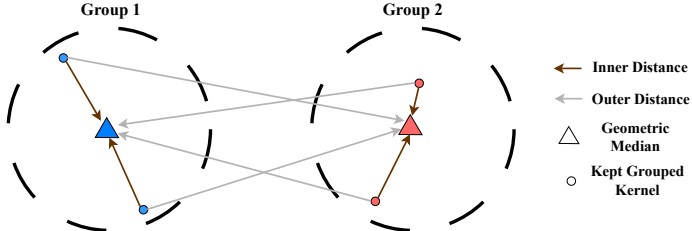

Figure 7: Visualization of LeanFlex-GKP Group Count Evaluation. We first compute the GM among retained grouped kernels and then calculate the inner and outer distance among them. After a normalization w.r.t. the group count, the one with the lowest average (Outer Distance − Inner Distance) is chosen; please refer to Appendix B.2 for details.

## 4 EXPERIMENTS

We evaluate the effectiveness of our method against 32 other densely structured pruning methods (Table 8) with coverage including BasicBlock (20/32/56/110) and BottleNeck ResNets (50/101) (He et al., 2016), VGG11/13/16 (Simonyan & Zisserman, 2015), DenseNet40 (Huang et al., 2017), and MobileNetV2 (Sandler et al., 2018). The datasets we used include CIFAR10/100 (Krizhevsky et al., 2009), Tiny-ImageNet (Wu et al., 2017), and ImageNet-1k (Deng et al., 2009). Please refer to Appendix D for full details on experiment settings.

### 4.1 RESULTS

Due to page limitations, here we only provide an abbreviated version of our experiments at Table 1. **We refer our readers to Table 9 to 16 in Appendix D for the full experiment results**, where we compared against 32 different structured pruning methods illustrated in Table 8 and evaluated our methods under 20 different settings specified in Table 5. **We also provide a series of ablation studies in Appendix C to facilitate an anatomical understanding of our proposed method.**

For all experiment results, **DA** represents if the method is data-agnostic (pruning can be done without access to data), **IP** indicates if a method is considered an iterative pruning method (utilizing a train-prune cycle), and **RB** reports recovery budget (in terms of epochs). All other reported criteria are in terms of %. **BA** and **Pruned** respectively report the unpruned (baseline) accuracy and the pruned accuracy. Methods marked with * are drawn from their original or (third-party) replicated publication; the rest are replicated by us to ensure a fair comparison (often with an identical baseline). Generally speaking, a method that is **DA** ✓, **IP** ✗, and demands a smaller **RB** is likely to be more user-friendly.

## 5 DISCUSSION AND CONCLUSION

We believe it is fair to conclude that our proposed method showcases SOTA-competitive (if not beyond) performance across comprehensive combinations of models and datasets. Out of all 20 reported results of LeanFlex-GKP, 17 of them showcased improvements after pruning (yet, no other compared method is able to provide positive ΔAcc under the three exception setups), suggesting our pruning method actually help on the generalization of the model should there be a reasonable setup.

We also note the compute (MACs) and memory (Params) reduction of our pruned models are almost always within 1% of their assigned pruning rates (e.g., see Table 15 and Table 16), which is a useful characteristic not found in many compared methods[2]. This supports one of the hassle-free claims we made in Section 2.3. Additionally, we would like to mention the combinations of BasicBlock ResNets with CIFAR10 — though being some of the most commonly evaluated combinations (Blalock et al., 2020) — are potentially getting saturated, as methods with significant performance gaps on more difficult model-dataset combinations tend to show little difference upon BasicBlock ResNets and CIFAR10.

In general, our empirical evaluation supports the efficacy of our flexible group count design as well as our goal of assembling a GKP method with only lightweight and low-dependency operations. Following the exposure of Park et al. (2023) for winning an *oral* recognition at AAAI 2023, our work serves as a more performant, efficient, and user-friendly advancement to the *grouped kernel pruning* granularity and can be of particular interest for both scholars of the pruning community or end users with practical application needs.

---

[2]This is evidenced by the many not-perfectly-aligned results in Table 15 and Table 16, where we tried to make all methods without the * mark — meaning we replicated such experiments under our controlled pipeline — aligned with the pruning rate in caption, but failed to do so in many scenarios.

Table 1: Abbreviated Experiment Results (please refer to Section 4.1 for header definitions)

| Method | DA | IP | RB | BA | Pruned | ΔAcc | ↓MACs | ↓Params |
|---|---|---|---|---|---|---|---|---|
| **VGG16 on CIFAR10** | | | MACs ≈ 313.4M | | Params ≈ 14.7M | | | |
| CC (Li et al., 2021) | ✗ | ✗ | 300 | 93.94 | 94.14 | ↑0.20 | 43.18 | - |
| GAL (Lin et al., 2019b) | ✗ | ✓ | 300 | 93.94 | 91.29 | ↓2.65 | 35.16 | 47.40 |
| HRank (Lin et al., 2020) | ✗ | ✓ | 300 | 93.94 | 93.57 | ↓0.37 | 32.28 | 40.82 |
| L1Norm (Li et al., 2017) | ✓ | ✗ | 300 | 93.94 | 92.88 | ↓1.06 | 42.71 | 37.85 |
| KPGP (Zhang et al., 2022b) | ✓ | ✗ | 300 | 94.27 | 94.17 | ↓0.13 | 43.15 | 43.59 |
| TMI-GKP (Zhong et al., 2022) | ✓ | ✗ | 300 | 93.94 | 94.07 | ↑0.10 | 25.00 | - |
| **LeanFlex-GKP (ours)** | ✓ | ✗ | 300 | 93.94 | **94.15** | ↑**0.21** | 43.15 | 43.59 |
| **ResNet110 on CIFAR10** | | | MACs ≈ 255.0M | | Params ≈ 1.73M | | | |
| TMI-GKP (Zhong et al., 2022) | ✓ | ✗ | 300 | 94.26 | 94.90 | ↑0.64 | 43.31 | 43.52 |
| L1Norm-B (Li et al., 2017) | ✓ | ✗ | 300 | 94.26 | 92.96 | ↓1.30 | 43.17 | 36.69 |
| CC (Li et al., 2021) | ✗ | ✗ | 300 | 94.26 | 94.31 | ↑0.05 | 44.54 | 39.47 |
| SFP (He et al., 2018a) | ✗ | ✓ | 300 | 94.26 | 94.44 | ↑0.18 | 43.42 | 43.52 |
| GAL (Lin et al., 2019a) | ✗ | ✓ | 300 | 94.26 | 93.42 | ↓0.84 | 29.14 | 31.37 |
| FPGM (He et al., 2019) | ✗ | ✓ | 300 | 94.26 | 94.18 | ↓0.08 | 43.39 | 43.52 |
| NPPM (Gao et al., 2021) | ✗ | ✗ | 300 | 94.26 | 94.16 | ↓0.10 | 42.46 | 35.19 |
| HRank (Lin et al., 2020) | ✗ | ✓ | 300 | 94.26 | 92.96 | ↓1.30 | 18.57 | 5.38 |
| DHP (Li et al., 2020) | ✗ | ✓ | 300 | 94.26 | 92.53 | ↓1.73 | 60.25 | 64.58 |
| LRF (Joo et al., 2021) | ✗ | ✗ | 300 | 94.26 | 94.49 | ↑0.23 | 43.37 | 42.30 |
| OTOv2 (Chen et al., 2023) | ✗ | ✓ | 300 | 94.26 | 91.58 | ↓2.68 | 37.83 | 42.44 |
| KPGP* (Zhang et al., 2022b) | ✓ | ✗ | 300 | 93.76 | 94.01 | ↑0.25 | 43.3 | 43.5 |
| **LeanFlex-GKP (ours)** | ✓ | ✗ | 300 | 94.26 | **94.92** | ↑**0.66** | 43.31 | 43.52 |
| **MobileNetV2 on CIFAR10** | | | MACs ≈ 98.768M | | Params ≈ 2.383M | | | |
| DCP* (Zhuang et al., 2018) | ✗ | - | 400 | 94.47 | 94.69 | ↑0.22 | 26.00 | - |
| SCOP* (Tang et al., 2020) | ✗ | - | 400 | 94.48 | 94.24 | ↓0.24 | 49.30 | - |
| WM* (Zhuang et al., 2018) | ✗ | - | 400 | 94.47 | 94.17 | ↓0.30 | 26.00 | - |
| DMC* (Gao et al., 2020) | ✗ | - | 160 | 94.23 | 94.49 | ↑0.26 | 40.00 | - |
| MDP* (Guo et al., 2020a) | ✗ | - | - | 95.02 | 95.14 | ↑0.12 | 28.71 | - |
| GDP* (Guo et al., 2021) | ✗ | - | 350 | 94.89 | 95.15 | ↑0.26 | 46.22 | - |
| ChipNet* (Tiwari et al., 2021) | ✗ | ✓ | 300 | 93.55 | 92.58 | ↓0.97 | 20.00 | - |
| **LeanFlex-GKP (ours)** | ✓ | ✗ | 300 | 93.87 | 94.30 | ↑**0.43** | 28.74 | 26.98 |
| **ResNet110 on CIFAR100** | | | MACs ≈ 255.001M | | Params ≈ 1.734M | | | |
| TMI-GKP (Zhong et al., 2022) | ✓ | ✗ | 300 | 72.99 | 72.79 | ↓0.20 | 43.31 | 43.37 |
| L1Norm-A (Li et al., 2017) | ✓ | ✗ | 300 | 73.20 | 69.85 | ↓3.35 | 43.74 | 44.41 |
| CC (Li et al., 2021) | ✗ | ✗ | 300 | 73.20 | 73.21 | ↑0.01 | 43.43 | 19.78 |
| NPPM (Gao et al., 2021) | ✗ | ✗ | 300 | 73.20 | 72.38 | ↓0.82 | 42.77 | 18.69 |
| LRF (Joo et al., 2021) | ✗ | ✗ | 300 | 73.20 | 73.58 | ↑0.38 | 43.38 | 42.16 |
| LCCL* (Dong et al., 2017) | ✗ | - | 300 | 72.79 | 70.78 | ↓2.01 | 31.3 | - |
| SFP* (He et al., 2018a) | ✗ | ✓ | 300 | 74.14 | 71.28 | ↓2.86 | 52.3 | - |
| FPGM* (He et al., 2019) | ✗ | ✓ | 300 | 74.14 | 72.55 | ↓1.59 | 52.3 | - |
| TAS* (Dong & Yang, 2019) | ✗ | ✗ | 300 | 75.06 | 73.16 | ↓1.90 | 52.6 | - |
| **LeanFlex-GKP (ours)** | ✓ | ✗ | 300 | 73.20 | **73.63** | ↑**0.43** | 43.31 | 43.36 |
| **ResNet56 on Tiny-ImageNet** | | | MACs ≈ 506.254M | | Params ≈ 0.865M | | | |
| TMI-GKP* (Zhong et al., 2022) | ✓ | ✗ | 300 | 55.59 | 51.48 | ↓4.11 | 37.05 | 36.76 |
| L1Norm-A (Li et al., 2017) | ✓ | ✗ | 300 | 56.13 | 55.41 | ↓0.72 | 35.51 | 32.14 |
| SFP (He et al., 2018a) | ✗ | ✓ | 300 | 56.13 | 53.65 | ↓2.48 | 33.96 | 35.38 |
| FPGM (He et al., 2019) | ✗ | ✓ | 300 | 56.13 | 54.14 | ↓1.99 | 33.53 | 34.68 |
| HRank (Lin et al., 2020) | ✗ | ✓ | 300 | 56.13 | 54.16 | ↓1.97 | 37.39 | 30.98 |
| GAL* (Lin et al., 2019b) | ✗ | ✓ | 100 | 56.55 | 55.87 | ↓0.68 | 52.00 | 32.00 |
| DHP* (Li et al., 2020) | ✗ | ✓ | 100 | 56.55 | 55.82 | ↓0.73 | 55.00 | 46.00 |
| 3D* (Wang et al., 2021) | ✗ | ✓ | 420 | 56.55 | **56.04** | ↓0.51 | 59.00 | 34.00 |
| Slimming* (Liu et al., 2017) | ✗ | ✗ | 100 | 56.55 | 52.45 | ↓4.10 | 53.00 | 54.00 |
| **LeanFlex-GKP (ours)** | ✓ | ✗ | 300 | 56.13 | 55.67 | ↓**0.46** | 37.05 | 36.76 |
| **ResNet50 on ImageNet-1K** | | | MACs ≈ 4122.828M | | Params ≈ 25.557M | | | |
| SFP* (He et al., 2018a) | ✗ | ✓ | 100 | 76.13 | 58.50 | ↓17.63 | 36.08 | 32.31 |
| FPGM* (He et al., 2019) | ✗ | ✓ | 100 | 76.13 | 75.04 | ↓1.09 | 35.93 | 28.36 |
| TMI-GKP* (Zhong et al., 2022) | ✓ | ✗ | 100 | 76.15 | 75.53 | ↓0.62 | 33.21 | 33.74 |
| ThiNet* (Luo et al., 2017) | ✗ | ✓ | 100 | 72.88 | 72.04 | ↓0.84 | 36.7 | - |
| OTOv2* (Chen et al., 2023) (post-train) | ✗ | ✓ | 120 | 76.13 | 75.38 | ↓0.75 | 37.70 | 26.58 |
| DOP* (Yang et al., 2022) | ✗ | ✗ | 120 | 76.47 | 74.29 | ↓2.18 | 60.00 | - |
| KPGP* (Zhang et al., 2022b) | ✓ | ✗ | | 76.15 | 75.50 | ↓0.65 | 33.7 | 33.2 |
| Layer-wise Proxy* (Elkerdawy et al., 2020) | ✗ | ✗ | - | 76.14 | 75.0 | ↓1.14 | 5.5 | - |
| **LeanFlex-GKP (ours)** | ✓ | ✗ | 100 | 76.13 | **75.62** | ↓**0.51** | 33.06 | 30.34 |

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

# A    RELATED WORKS

**Structured pruning.**    As discussed in Section 1.1 and illustrated Figure 1, structured pruning is a family of pruning methods that removes model components in groups, where the majority of methods are capable of delivering densely structured pruned models for immediate efficiency benefits. Below, we briefly introduce some important aspects of structured pruning and refer our reader to comprehensive survey work like Blalock et al. (2020) and He & Xiao (2023) for details.

Prior papers on densely structured pruning have carried out such pruning operations on (or determined by) filters, channels, layers, inputs, or combinations of the above (Li et al., 2017; Dong et al., 2017; Anwar et al., 2017; Wang et al., 2021; Li et al., 2021; Fang et al., 2023; Li et al., 2020; He et al., 2019; Lin et al., 2019a; 2020; Li et al., 2017; Joo et al., 2021; Gao et al., 2021; Chen et al., 2023; Wang et al., 2020; He et al., 2018a; Liu et al., 2017; Luo et al., 2017; Zhong et al., 2022; Zhuang et al., 2018; Gao et al., 2020; Guo et al., 2021; Dong et al., 2017; Guo et al., 2020a; Tang et al., 2020; Dong & Yang, 2019; Wang et al., 2019; Zhao et al., 2023; He et al., 2018b; Tiwari et al., 2021; Yang et al., 2022; Zhang et al., 2022b; Zhuang et al., 2018; Liebenwein et al., 2020; Elkerdawy et al., 2020; Guo et al., 2020c). Scholars often refer to this as the "**granularity**" or simply the "type" **of a pruning method**. Among them, filter/channel pruning is considered the most popular structured pruning granularity as it can gain immediate efficiency benefits after channel removal.

Outside the pruning granularity, the **pipeline of a pruning method** plays another major role in deciding which method to adopt. Pruning operations can conducted from-scratch (e.g., OTOv2 by Chen et al. (2023)), during training (e.g., Roy et al. (2020)), or post-train (e.g., FPGM by He et al. (2019)). Many pruning methods also require intervention (or access to information) at an earlier stage, then conduct actual pruning at a later stage. E.g., DMCP by Guo et al. (2020c) further adjusts weights of a pruned model before pruning, and TMI-GKP by Zhong et al. (2022) requires access to the model training checkpoint to guide its pruning operations. Most pruning methods follow a *train - prune - fine-tune/retrain* pipeline, as the trained model provides a good starting and reference point for pruning and evaluating. However, this pipeline suffers the natural drawback of having to both train the unpruned baseline model and fine-tune/retrain the pruned model, where from-scratch methods or fine-tuning free methods like Narshana et al. (2023) may effectively avoid such compute cost, though often with a trade-off of delivering lower accuracy retention.

Further, the **scheduling of a pruning method** drastically affects the efficiency and adaptability of a method. Most pruning methods can be roughly categorized into one-shot or iterative pruning. The former prunes all redundant model components all at once; the latter, as its name implies, conducts pruning gradually with weight updates between two pruning operations. One-shot pruning is often considered easier to deploy and more efficient to run, though iterative pruning is generally more performant on accuracy retention.

On the note of adaptability, the **data dependency of a pruning method** potentially plays another vital role in the adaptability of a method. With *data-agnostic* pruning methods do not require data access to determine what to prune, and *data-dependent* or *data-informed* methods do the opposite.

Other **implementations details** such as *hard* or *soft pruning* (whether the pruned components are entirely removed from the model to yield a pruned model with reduced dimension, or just zero-masked), reduction control and estimation (if one can reliably control and predict the memory and compute requirements of a pruned model before conducting the actual pruning), and hyperparameter tuning pressure (whether the method has a lot of hyperparameters to adjust, or if the method has a way to tune them automatically, like AAP by Zhao et al. (2023) and AMC by He et al. (2018b)) also have their influences, especially on the user-friendliness aspect of a method.

Our method is a one-shot, post-train, data-agnostic, hard-pruning method with only one tunable hyperparameter (which is primarily determined by the layer dimension; see Appendix C.2 for details). Yet, one may adjust the pruning rate setting of our method to reliably control and predict the *MACs* and *Params* reduction of the pruned model (see the relation between pruning rate and pruned model size in experiments like Table 15 and 16). These characteristics put our method on top of the efficiency, adaptability, and user-friendliness lists over many other pruning methods under the context of densely structured CNN pruning.

**Grouped kernel pruning.** Grouped kernel pruning (GKP) is a special type of densely structured pruning granularity that is able to offer finer pruning freedom than filter or channel pruning methods. As demonstrated in Figure 2, GKP is, in essence, a combination of pruning a stack of kernels under the same filter group (a.k.a. grouped kernels) and reconstructing the remaining model to a grouped convolution format (Krizhevsky, 2014). Granted, grouped convolution itself is proposed with efficiency motivations, and we may say that this granularity — outside a pruning context — naturally exists with the debut of AlexNet (Krizhevsky et al., 2012).

The combination of densely structured pruning with grouped convolution is first introduced in Yu et al. (2017), but the method, unfortunately, did not attract significant attention from the community. One potential reason for this is due to the high complexity of its pruning procedure: feature maps-dependent iterative pruning plus fine-tuning with knowledge distillation. Other reasons include not performing a comprehensive ablation study to track its contributions, as well as a lack of comparable experiments to modern pruning arts. Another work, Guo et al. (2020b), also explores structured pruning with grouped convolution, but it still introduces sparsity to its pruned model with its zero-padded implementation to support unequal group sizes.

A refined procedure at the intersection of densely structured pruning and grouped convolution is brought by Zhong et al. (2022) as TMI-GKP, where they stipulate a three-stage procedure consisting of 1) filter grouping, 2) grouped kernel pruning, and 3) grouped convolution reconstruction – as well as coining it with the term *"grouped kernel pruning."* The GKP procedure and nomenclature have since been adopted by TMI-GKP's concurrent, related, and follow-up works like Zhang et al. (2022a); Park et al. (2023); He & Xiao (2023).

# B    EXTENDED PROPOSED METHOD

## B.1    FORMAL DEFINITION OF $L_2$ & GEOMETRIC MEDIAN-BASED GROUPED KERNEL PRUNING

To better illustrate the formal definition of our purposed method, let $W$ be the set containing all filters $f$ in a certain convolutional layer of a CNN:

$$W : \{f_1, f_2, \ldots, f_n\},$$

where $n$ is the number of filters in that layer, a.k.a. the `out_channels`. The filter grouping procedure is then applied on such $f_1$ to $f_n$.

The numbers of desire grouped filters (a.k.a. the `Conv2d(groups)` or group count) are chosen from *candidate group count* list (see Table 5), where one of them is ultimately selected after the *post-prune group count evaluation* (see Section 3.4).

For simplicity, let's assume that the current *candidate group count* is $m$ (the number of filters $n$ must be divisible by $m$ because in our implementation every group in the same layer has exactly the same number of filters to be able to reconstruct to a grouped convolution format), and let $G$ be the set containing all filter groups $g$ after KPP-aware filter clustering is applied on all the filters in $W$ (Figure 5):

$$W : \{f_1, f_2, \ldots, f_n\} \xrightarrow{\text{KPP-aware Filter Grouping}} G : \{g_1, g_2, \ldots, g_m\},$$

where $g_i$ is the $i_{th}$ grouped filters in $G$.

Suppose there are $\lambda$ number of kernels in a filter (a.k.a. `in_channel`), we shall have $\lambda$ grouped kernels ($gk$) in each filter group $g$:

$$g : \{gk_1, gk_2, \ldots, gk_\lambda\}.$$

We then compute the geometric median $\tau$ of each filter group $g$ among all grouped kernels ($gk$) in $g$:

$$G : \{g_1, g_2, \ldots, g_m\} \xrightarrow{\text{Compute Geometric Median}} \{\tau_1, \tau_2, \ldots, \tau_m\}.$$

After we get each group's geometric median, we start to prune grouped kernels $gk$ in each filer group $g$. Inside each filter group $g$, we compute the L2-norm of each grouped kernels $gk$ in $g$ and add them to list $Q$:

$$g : \{gk_1, gk_2, \ldots, gk_\lambda\} \xrightarrow{\text{Compute L2-Norm}} Q : \{q_1, q_2, \ldots, q_\lambda\}, \text{where } q_i = \|gk_i\|_2.$$

Also, we compute the euclidean distance of each grouped kernels $gk$ to its group's geometric median $\tau$ and add them to list $D$:

$$g : \{gk_1, gk_2, \ldots, gk_\lambda\} \xrightarrow{\text{gk's Distance to } \tau} D : \{d_1, d_2, \ldots, d_\lambda\}, \text{where } d_i = \text{Euclidean}(gk_i, \tau).$$

Then we do Min-Max Normalization on list $Q$ and $D$ separately:

$$Q : \{q_1, q_2, \ldots, q_\lambda\} \xrightarrow{\text{Min-Max Normalization}} Q' : \{q_1', q_2', \ldots, q_\lambda'\},$$

such that

$$q_i' = \frac{q_i - \min(Q)}{\max(Q) - \min(Q)}.$$

Using the same equation above, we conduct the same Min-Max Normalization upon list $D$:

$$D : \{d_1, d_2, \ldots, d_\lambda\} \xrightarrow{\text{Min-Max Normalization}} D' : \{d_1', d_2', \ldots, d_\lambda'\}.$$

We then calculate the importance score $I$ of each grouped kernels ($gk$) (Figure 6):

$$g : \{gk_1, gk_2, \ldots, gk_\lambda\} \xrightarrow{\text{calculate importance score}} I : \{I_1, I_2, \ldots, I_\lambda\}, \text{where } I_i = q_i' + d_i'.$$

Finally, assume that the pruning rate (ratio of grouped kernels to be pruned) is $pr$, we preserve $1 - pr$ ratio of grouped kernels $gk$ in group $g$ with higher importance score $I$. So with $\lambda$ number of $gk$ and preserve rate $1 - pr$, we will have $\lambda(1 - pr)$ numbers of preserved grouped kernels $gk$ after pruning:

$$\underbrace{g : \{gk_1, gk_2, \ldots, gk_\lambda\}}_{\lambda \text{ numbers of } gk} \xrightarrow{\text{pruning}} \underbrace{g^* : \{gk_1^*, gk_2^*, \ldots, gk_\lambda^*\}}_{\lambda(1 - pr) \text{ numbers of } gk}.$$

So in general, for each *candidate group count* $m$, we will have GKP result $G^*$:

$$\underbrace{W : \{f_1, f_2, \ldots, f_n\} \to G : \{g_1, g_2, \ldots, g_m\}}_{\text{filter grouping stage}} \to \underbrace{G^* : \{g_1^*, g_2^*, \ldots, g_m^*\}}_{gk \text{ pruning stage}}.$$

For demonstration, suppose we have three *candidate group count* $[m_1, m_2, m_3]$, we will have three pruning results $G_1^*, G_2^*, G_3^*$. Then we evaluate these three pruning results via *post-prune group count evaluation* (Section 3.4). Finally, based on *post-prune group count evaluation*, the optimal pruning result will be transformed into densely structured group convolution layer.

## B.2 FORMAL DEFINITION OF POST-PRUNE GROUP COUNT EVALUATION

For simplicity, let's assume that the current *candidate group count* is $m$ ($m$ number of groups in a layer), and suppose there are $\lambda$ number of kernels in a filter. After grouped kernel pruning (Section B.1), we can get pruned layer:

$$G^* : \{g_1^*, g_2^*, \ldots, g_m^*\}.$$

Then we may compute the preserved grouped kernels geometric median $\tau^*$ for each pruned group $g^*$:

$$G^* : \{g_1^*, g_2^*, \ldots, g_m^*\} \xrightarrow{\text{Compute Geometric Median}} \{\tau_1^*, \tau_2^*, \ldots, \tau_m^*\},$$

where $\tau_i^*$ is the **geometric median of preserved gk** (different from Section 3.3, which are the geometric median for all $gk$ including those being pruned). Let $|\,|$ be an operator that calculate cardinality in terms of grouped kernels. For group $g_i^*$ with geometric median $\tau_i^*$:

$$A(g_i^*) = \frac{1}{|g_i^*|} \sum_{gk \in g_i^*} \text{Euclidean}(gk, \tau_i^*),$$

where $A(g_i^*)$ is the average euclidean distance between $gk$ of $i^{th}$ group and its geometric median (intra-group similarity).

$$B(g_i^*) = \frac{1}{|G^* - g_i^*|} \sum_{gk \in G^* - g_i^*} \text{Euclidean}(gk, \tau_i^*),$$

where $B(g_i^*)$ is the average euclidean distance between $i^{th}$ group's geometric median and all $gk$ that are not in $i^{th}$ group (inter-group distinctions). Now, we may calculate the group count evaluation score $S$ upon the pruning result of each candidate group count (see Figure 7) as:

$$S = \gamma \sum_{i=1}^{m} B(g_i^*) - A(g_i^*), \text{where } \gamma = \frac{\lambda}{m}. \tag{1}$$

Since we have *candidate group count* list with different group counts (see Figure 5), during *post-prune group count evaluation*, each group count will provide a unique score $S$ using Formular 1.

For demonstration, following the example in the last paragraph of Section B.1, suppose we have three *candidate group counts* $[m_1, m_2, m_3]$, we will have three pruning results $G_1^*, G_2^*, G_3^*$. Using Equation 1, we will have three *group count evaluation scores* $S_1, S_2, S_3$. So we only need to choose the pruning result $G^*$ among $G_1^*, G_2^*, G_3^*$ with largest $S$ value in $S_1, S_2, S_3$.

A large value of $S$ implies a pruned layer with better intra-group similarity and inter-group distinctions. Utilizing this $S$ evaluation metric, we are able to leverage the joint optimization effect of grouping and pruning on different group count settings and, therefore, obtain a better pruning result for the layer-in-question.

### B.3  PSEUDO CODE FOR THE PROPOSED METHOD

---
**Algorithm 1** General Procedure of LeanFlex-GKP on a Single Convolutional Layer

---
    **Input:** Candidate Group Count Queue $Q$     ▷ candidate `Conv2d(groups)` like `[2,4,8]`
1:  **Initialize:** Empty List $P$     ▷ storage of pruning strategies w.r.t. each group count candidate
2:  **for** $q \in Q$ **do**     ▷ looping though all group count candidates
3:      Conduct $k$-Means$^{++}$ clustering on filters to get $q$ amount of centers
4:      $CS \leftarrow q$ candidate sequences of centers generated by multiple restarts     ▷ see Figure 5
5:      **Initialize:** Empty List $FG$     ▷ to store filter grouping results
6:      **for** $s \in CS$ **do**
7:        Get filter grouping result $fg_s$ for the $s$ center sequence, as illustrated in Figure 5(c)
8:        $FG$.append($fg_s$)
9:      Determine the optimal filter grouping result $fg_{\text{opt}}$ with least intra-group distance (Figure 5)
10:     **for** filter group $g \in fg_{\text{opt}}$ **do**     ▷ Prune grouped kernels inside each filter group
11:       $g_{\text{pruned}}^q \leftarrow$ Prune $g$ w.r.t. the $L_2$ & geometric median-based method stipulated in Figure 6
12:     $P$.append($g_{\text{pruned}}^q$)
13: Determine the best $g_{\text{opt}} \in P$ according to Appendix B.2     ▷ illustrated in Figure 7
14: **return** Pruned convolutional layer $g_{\text{opt}}$ and its corresponding group count $q_{\text{opt}}$

---

# C  ABLATION STUDIES

In company with our main experiment results showcased in Section 4 and Appendix D, we provide ablation studies of our proposed LeanFlex-GKP method from different perspective-of-interests,

## C.1  ON DIFFERENT PROCEDURAL RECIPES

In this section, we conduct ablation studies on our proposed methods by evaluating the contribution of our proposed algorithmic components. We utilized BasicBlock ResNets (He et al., 2016) on the CIFAR10 dataset (Krizhevsky et al., 2009) for their lightweightness. All comparisons are done so with the pruning rate being $43.75\%$ (meaning $43.75\%$ of the original model is removed; please refer to the LeanFlex-GKP reports in Table 16 for the exact reduction status).

In Table 2, we try to investigate the influence of different grouping approaches under our Flexible Group Count pipeline, where our proposed filter grouping method, **KPP-aware filter grouping** (Section 3.2) has the optimal empirical results.

Table 2: Ablation Study on Different Filter Grouping Approaches

| Method Name | ResNet32 | ResNet56 | ResNet110 |
|---|---|---|---|
| FGC + RandomGroup + GM&L2 | 92.81 | 93.72 | 94.67 |
| FGC + No Restart KPP-aware Filter Grouping + GM&L2 | **93.04** | 93.76 | 94.61 |
| FGC + Equal-size KPP + GM&L2 | 92.52 | 93.77 | - |
| **FGC + KPP-aware Filter Grouping + GM&L2 (ours)** | 93.01 | **94.00** | **94.92** |

In Table 3, we try to evaluate the validity of our proposed $L_2$ **& geometric median-based grouped kernel pruning** method in comparison with other pruning strategies (while under our flexible group count pipeline). It is observed that our flexible group count pipeline is best working with the GM&L2 grouped kernel pruning method we proposed in Section 3.3.

Table 3: Ablation Study on Different Grouped Kernel Pruning Approaches

| Method Name | ResNet32 | ResNet56 |
|---|---|---|
| FGC + KPP-aware Filter Grouping + TMI (Greedy) | 92.84 | 93.58 |
| FGC + KPP-aware Filter Grouping + Distance to GM | 92.92 | 93.73 |
| FGC + KPP-aware Filter Grouping + L2Norm | 92.99 | 93.61 |
| **FGC + KPP-aware Filter Grouping + GM&L2 (ours)** | **93.01** | **94.00** |

In Table 4, we showcase the power of adding flexible group count into our GKP procedure as stipulated in **post-prune group count evaluation** (Section 3.4), it is observed that the `Conv2d(groups)` decided by our evaluation design has better performance than other constant or dynamic group count settings. Specifically, the "Eight Groups" strategy is utilized in TMI-GKP (Zhong et al., 2022), and the "Random Groups Reassign" means to randomly re-distribute the optimal group counts (deemed by our post-prune group count evaluation method) across different layers; granted such reassignment is legal in term of layer dimensions.

Table 4: Ablation Study on Different Group Count Settings

| Method Name | ResNet32 | ResNet56 | ResNet110 |
|---|---|---|---|
| Eight Groups (TMI's setting) + KPP-aware Filter Grouping + GM&L2 | 92.86 | 93.82 | 94.72 |
| Max Groups + KPP-aware Filter Grouping + GM&L2 | 92.88 | 93.82 | 94.72 |
| Random Groups Reassign + KPP-aware Filter Grouping + GM&L2 | 92.75 | 93.89 | 94.43 |
| **FGC + KPP-aware Filter Grouping + GM&L2 (ours)** | **93.01** | **94.00** | **94.92** |

## C.2  ON DIFFERENT HYPERPARAMETER SETTINGS

We illustrate the hyperparameter settings for all reported LeanFlex-GKP experiments as Table 5. The only tunable hyperparameter for LeanFlex-GKP is Candidate Group Counts, i.e., a set of `Conv2D(groups)` setting considered. Granted most CNN architectures have different dimensions across their convolutional layer, this setting should be adjusted subject to subject to the layer's `out_channels`. In our case, we basically checkout what is the largest `Conv2D(groups)`

applicable to a particular convolutional layer, then generate the rest of group count candidates by reducing it by half.

Table 5: Hyperparameter Settings for LeanFlex-GKP's Reported Results. **PR** stands for pruning rate, **Budget** represents the *train - fine-tune* budget in terms of number of epochs, **BS** implies batch sizes, and **Candidate Group Counts** indicate the different `Conv2D(groups)` settings considered. We provide settings in `torch` style code snippets

| Model | Dataset | PR | Budget | BS | Optimizer & Learning Rate | Candidate Group Counts |
|---|---|---|---|---|---|---|
| ResNet20 ResNet32 ResNet56 ResNet110 | CIFAR10 | 43.75% | 300 - 300 | 64 | SGD(lr=0.01, momentum=0.9, weight_decay=5e-4) StepLR(step_size=100, gamma=0.1) | S2 S1 S2 S1 |
| ResNet32 ResNet56 ResNet110 | CIFAR10 | 62.5% | 300 - 300 | 64 | SGD(lr=0.01, momentum=0.9, weight_decay=5e-4) StepLR(step_size=100, gamma=0.1) | S1 S2 S1 |
| ResNet56 ResNet110 | CIFAR100 | 43.75% | 200-300 | 64 | SGD(lr=0.01, momentum=0.9, weight_decay=5e-4) StepLR(step_size=100, gamma=0.1) | S2 S2 |
| ResNet56 | Tiny-ImageNet | 37.5% | 100-300 | 64 | SGD(lr=0.01, momentum=0.9, weight_decay=5e-4) MultiStepLR(milestones=[80, 90], gamma=0.1) | S2 |
| ResNet101 | Tiny-ImageNet | 43.75% | 300-20 | 256 | SGD(lr=0.1, momentum=0.9, weight_decay=1e-4) StepLR(step_size=5, gamma=0.1) | [8,16,32]x1 [8,16,32,64]x3 |
| ResNet50 | ImageNet | 66.00% | Pretrained-100 | 256 | SGD(lr=0.01, momentum=0.9, weight_decay=1e-4) StepLR(step_size=30, gamma=0.1) | All layers [4,8,16,32] |
| VGG11 | CIFAR10 | 43.75% | 300-300 | 64 | SGD(lr=0.01, momentum=0.9, weight_decay=5e-4) StepLR(step_size=100, gamma=0.1) | [8,16,32]x1 [16,32,64]x2 [32,64,128]x4 |
| VGG13 | CIFAR10 | 43.75% | 300-300 | 64 | SGD(lr=0.01, momentum=0.9, weight_decay=5e-4) StepLR(step_size=100, gamma=0.1) | [4,8,16]x2 [8,16,32]x2 [16,32,64]x2 [32,64,128]x3 |
| VGG16 | CIFAR10 | 43.75% | 300-300 | 64 | SGD(lr=0.01, momentum=0.9, weight_decay=5e-4) StepLR(step_size=100, gamma=0.1) | [4,8,16]x2 [8,16,32]x3 [16,32,64]x3 [32,64,128]x4 |
| DenseNet40 | CIFAR10 | 33.33% 50.00% 66.67% | Pretrained-300 | 64 | SGD(lr=0.01, momentum=0.9, weight_decay=1e-4) MultiStepLR(milestones=[150, 225], gamma=0.1) | [4,6,12] [7,8,12,14,21,24] [8,12,13,24] |
| DenseNet40 | CIFAR10 | 52.00% | 400-300 | 64 | SGD(lr=0.01, momentum=0.9, weight_decay=1e-4) MultiStepLR(milestones=[150, 225], gamma=0.1) | [4,6,12] [7,8,12,14,21,24] [8,12,13,24] |
| MobileNetV2 | CIFAR10 | 25.00% | 300-300 | 64 | SGD(lr=0.001, momentum=0.9, weight_decay=5e-4) StepLR(step_size=100, gamma=0.1) | [2,4,8] [4,8,12,24] [4,8,16] [8,16,32,64] [8,16,32] |

Note in Table 5 above we have S1 and S2 as candidate group count settings for BasicBlock ResNets on CIFAR10. In such cases, S1 stands for [4,8,16], [8,16,32], [16,32,64], where S2 stands for [8,16], [8,16,32], [8,16,32,64]. Table 6 reports the evaluation results when the models are grouped/pruned according to different candidate group count settings. Under most evaluated setups, our method's performance is similar between the two settings.

Table 6: Basicblock ResNets on CIFAR10 when pruned according to setting S1 and S2.

| Model | S1: pr $= 43.75\%$ | S2: pr $= 43.75\%$ | S1: pr $= 62.50\%$ | S2: pr $= 62.50\%$ |
|---|---|---|---|---|
| ResNet20 | 92.14 | **92.49** | - | - |
| ResNet32 | **93.01** | 92.96 | **92.40** | 92.07 |
| ResNet56 | 93.93 | **94.00** | 93.32 | **93.54** |
| ResNet110 | **94.92** | 94.54 | **94.35** | 94.34 |

## C.3 ON WALL-CLOCK SPEEDUP

We additionally investigate the wall-clock runtime of our proposed method at Table 7 regarding its pruning procedure (time required to obtain a ready-to-fine-tune pruned model in grouped convolution format).

Table 7: Wall-clock Runtime Comparison between LeanFlex-GKP (Ours) and TMI-GKP (Zhong et al., 2022)'s Pruning Procedure

| Method | ResNet32 | ResNet56 | ResNet110 |
|---|---|---|---|
| TMI-GKP (Zhong et al., 2022) | 1h 20m 10s | 2h 36m 22s | 5h 30m 18s |
| **LeanFlex-GKP (ours)** | 10m 56s | 21m 32s | 44m 15s |

Our method provides a massive speed advantage over TMI-GKP while both being post-train, one-shot, and data-agnostic.

# D  EXTENDED EXPERIMENT RESULTS

Our proposed method, LeanFlex-GKP, follows the classic *train - prune - fine-tune* pipeline under a data-agnostic setting. This implies all model components are pruned all at once prior to fine-tuning, without having access to the training or fine-tuning data. Our method is implemented in a hard pruning fashion, which means the pruned model for fine-tuning is already compressed. We refer our readers to Table 5 for specific experiment details such as epoch budget and hyperparameter settings, as we have thereat documented detailed experiment settings for all 20 reported results of LeanFlex-GKP.

As introduced in Section 4, we evaluate the effectiveness of our method against many other densely structured pruning methods on ResNet20/32/56/110 with the BasicBlock, ResNet50/101 with the BottleNeck implementation (He et al., 2016), VGG11/13/16 (Simonyan & Zisserman, 2015), DenseNet40 (Huang et al., 2017), and MobileNetV2 (Sandler et al., 2018). The datasets we used include CIFAR10/100 (Krizhevsky et al., 2009), Tiny-ImageNet (Wu et al., 2017), and ImageNet-1k (Deng et al., 2009).

## D.1  COMPARED METHODS

Our methods is compared against 32 different pruning methods as illustrated in Table 8. Where notions like C/F/GK/K/L/R in the **Granularity** column respectively represent Channel/Filter/Grouped Kernel/Kernel/Layer/Resolution pruning. **Procedure** indicates if the pruned model is generated iteratively (requires weight update between conducting the first pruning act and having the fully pruned model) or in a one-shot manner (pruned all at once without weight update in between). **Zero-Masked?** column investigates if a model is hard pruned (no zero-masked weight) before fine-tuning.

Table 8: Overview of Compered Methods.

| Method | Venue | Granularity | Procedure | Zero-Masked? |
|---|---|---|---|---|
| 3D (Wang et al., 2021) | ICML | F&L&R | Iterative | - |
| CC (Li et al., 2021) | CVPR | C | One-shot | N |
| DepGraph (Fang et al., 2023) | CVPR | C | One-shot | N |
| DHP (Li et al., 2020) | ECCV | F | Iterative (from-scratch) | Y |
| FPGM (He et al., 2019) | CVPR | F | Iterative | Y |
| GAL (Lin et al., 2019a) | CVPR | F | Iterative | Y |
| HRank (Lin et al., 2020) | CVPR | F | Iterative | Y |
| L1Norm (Li et al., 2017) | ICLR | F | One-shot | N |
| LRF (Joo et al., 2021) | AAAI | C | One-shot | N |
| NPPM (Gao et al., 2021) | CVPR | C | One-shot | N |
| OTOv2 (Chen et al., 2023) | ICLR | F | Iterative (from-scratch) | Y |
| PScratch (Wang et al., 2020) | AAAI | C | One-shot (from-scratch) | - |
| SFP (He et al., 2018a) | IJCAI | F | Iterative | Y |
| Slimming (Liu et al., 2017) | ICCV | C | One-shot | N |
| ThiNet (Luo et al., 2017) | ICCV | F | One-shot | - |
| TMI-GKP (Zhong et al., 2022) | ICLR | GK | One-shot | N |
| **LeanFlex-GKP (Ours)** | - | GK | One-shot | N |
| EigenDamage (Wang et al., 2019) | ICML | C | Iterative | N |
| AAP (Zhao et al., 2023) | AISTATS | F | Iterative | - |
| AMC (He et al., 2018b) | ECCV | C | - | N |
| ChipNet (Tiwari et al., 2021) | ICLR | C | Iterative | N |
| PFP (Liebenwein et al., 2020) | ICLR | F | - | - |
| GDP (Guo et al., 2021) | ICCV | C | One-shot | - |
| DOP (Yang et al., 2022) | BMVC | C | One-shot | Y |
| KPGP (Zhang et al., 2022b) | APIN | GK | One-shot | N |
| WM (Zhuang et al., 2018) | NeurIPS | C | Iterative | - |
| Layer-wise Proxy (Elkerdawy et al., 2020) | IEEE ICIP | L | One-shot | N |
| DCP (Zhuang et al., 2018) | NeurIPS | C | Iterative | - |
| DMC (Gao et al., 2020) | CVPR | C | Iterative | - |
| LCCL (Dong et al., 2017) | CVPR | K | Iterative | Y |
| MDP (Guo et al., 2020a) | CVPR | C | - | - |
| SCOP (Tang et al., 2020) | NeurIPS | F | - | - |
| TAS (Dong & Yang, 2019) | NeurIPS | C&L | Iterative (NAS) | - |

### D.2 FULL EXPERIMENT RESULTS

The terms and notations utilized in the following experiment results follow the definitions defined in Section 4.1: **DA** represents if the method is data-agnostic (pruning can be done without access to data), **IP** indicates if a method is considered an iterative pruning method (utilizing a train-prune cycle), and **RB** reports recovery budget (in terms of epochs). All other reported criteria are in terms of %. **BA** and **Pruned** respectively report the unpruned (baseline) accuracy and the pruned accuracy. Methods marked with * are drawn from their original or (third-party) replicated publication; the rest are replicated by us to ensure a fair comparison. Generally speaking, a method that is **DA** ✓, **IP** ✗, and demands a smaller **RB** is likely to be more user-friendly.

Table 9: Results of ResNet50 Model on ImageNet-1K Dataset. Results in **bold red** indicate being the second best among comparisons.

| Method | DA | IP | RB | BA | Pruned | △Acc | ↓MACs | ↓Params |
|---|---|---|---|---|---|---|---|---|
| **ResNet50 on ImageNet-1K** | | | | MACs ≈ 4122.828M | | Params ≈ 25.557M | | |
| SFP* (He et al., 2018a) | ✗ | ✓ | 100 | 76.13 | 58.50 | ↓17.63 | 36.08 | 32.31 |
| FPGM* (He et al., 2019) | ✗ | ✓ | 100 | 76.13 | 75.04 | ↓1.09 | 35.93 | 28.36 |
| TMI-GKP* (Zhong et al., 2022) | ✓ | ✗ | 100 | 76.15 | 75.53 | ↓0.62 | 33.21 | 33.74 |
| PScratch* (Wang et al., 2020) | ✗ | ✓ | ≈ 409 | 77.20 | **76.70** | ↓**0.50** | 29.80 | 26.80 |
| ThiNet* (Luo et al., 2017) | ✗ | ✓ | 100 | 72.88 | 72.04 | ↓0.84 | 36.7 | - |
| OTOv2* (Chen et al., 2023) (post-train) | ✗ | ✓ | 120 | 76.13 | 75.38 | ↓0.75 | 37.70 | 26.58 |
| DOP* (Yang et al., 2022) | ✗ | ✗ | 120 | 76.47 | 74.29 | ↓2.18 | 60.00 | - |
| Layer-wise Proxy* (Elkerdawy et al., 2020) | ✗ | ✗ | - | 76.14 | 75.0 | ↓1.14 | 5.5 | - |
| KPGP* (Zhang et al., 2022b) | ✓ | ✗ | | 76.15 | 75.50 | ↓0.65 | 33.7 | 33.2 |
| **LeanFlex-GKP (ours)** | ✓ | ✗ | 100 | 76.13 | **75.62** | ↓**0.51** | 33.06 | 30.34 |

Table 10: Results of ResNet56/101 Model on Tiny-ImageNet Dataset

| Method | DA | IP | RB | BA | Pruned | △Acc | ↓MACs | ↓Params |
|---|---|---|---|---|---|---|---|---|
| **ResNet56 on Tiny-ImageNet** | | | | MACs ≈ 506.254M | | Params ≈ 0.865M | | |
| TMI-GKP* (Zhong et al., 2022) | ✓ | ✗ | 300 | 55.59 | 51.48 | ↓4.11 | 37.05 | 36.76 |
| L1Norm-A (Li et al., 2017) | ✓ | ✗ | 300 | 56.13 | 55.41 | ↓0.72 | 35.51 | 32.14 |
| L1Norm-B (Li et al., 2017) | ✓ | ✗ | 300 | 56.13 | 55.21 | ↓0.92 | 36.43 | 41.04 |
| SFP (He et al., 2018a) | ✗ | ✓ | 300 | 56.13 | 53.65 | ↓2.48 | 33.96 | 35.38 |
| FPGM (He et al., 2019) | ✗ | ✓ | 300 | 56.13 | 54.14 | ↓1.99 | 33.53 | 34.68 |
| HRank (Lin et al., 2020) | ✗ | ✓ | 300 | 56.13 | 54.16 | ↓1.97 | 37.39 | 30.98 |
| GAL* (Lin et al., 2019b) | ✗ | ✓ | 100 | 56.55 | 55.87 | ↓0.68 | 52.00 | 32.00 |
| DHP* (Li et al., 2020) | ✗ | ✓ | 100 | 56.55 | 55.82 | ↓0.73 | 55.00 | 46.00 |
| 3D* (Wang et al., 2021) | ✗ | ✓ | 420 | 56.55 | **56.04** | ↓0.51 | 59.00 | 34.00 |
| Slimming* (Liu et al., 2017) | ✗ | ✗ | 100 | 56.55 | 52.45 | ↓4.10 | 53.00 | 54.00 |
| **LeanFlex-GKP (ours)** | ✓ | ✗ | 300 | 56.13 | **55.67** | ↓**0.46** | 37.05 | 36.76 |
| **ResNet101 on Tiny-ImageNet** | | | | MACs ≈ 10081.092M | | Params ≈ 42.902M | | |
| TMI-GKP* (Zhong et al., 2022) | ✓ | ✗ | 300 | 65.51 | 66.89 | ↑1.38 | 43.25 | 43.53 |
| GAL* (Lin et al., 2019b) | ✗ | ✓ | 100 | 64.83 | 64.33 | ↓0.50 | 76.00 | 45.00 |
| DHP* (Li et al., 2020) | ✗ | ✓ | 100 | 64.83 | 64.82 | ↓0.01 | - | 50.00 |
| 3D* (Wang et al., 2021) | ✗ | ✓ | 420 | 64.83 | 65.27 | ↑0.44 | - | 51.00 |
| Slimming* (Liu et al., 2017) | ✗ | ✗ | 100 | 64.83 | 63.47 | ↓1.36 | - | 75.00 |
| **LeanFlex-GKP (ours)** | ✓ | ✗ | 20 | 65.51 | **68.46** | ↑**2.95** | 43.25 | 43.53 |

Table 11: Results of MobileNetV2 Model on CIFAR10 Dataset

| Method | DA | IP | RB | BA | Pruned | △Acc | ↓MACs | ↓Params |
|---|---|---|---|---|---|---|---|---|
| **MobileNetV2 on CIFAR10** | | | | MACs ≈ 98.768M | | Params ≈ 2.383M | | |
| DCP* (Zhuang et al., 2018) | ✗ | - | 400 | 94.47 | 94.69 | ↑0.22 | 26.00 | - |
| SCOP* (Tang et al., 2020) | ✗ | - | 400 | 94.48 | 94.24 | ↓0.24 | 49.30 | - |
| WM* (Zhuang et al., 2018) | ✗ | - | 400 | 94.47 | 94.17 | ↓0.30 | 26.00 | - |
| DMC* (Gao et al., 2020) | ✗ | - | 160 | 94.23 | 94.49 | ↑0.26 | 40.00 | - |
| MDP* (Guo et al., 2020a) | ✗ | - | - | 95.02 | 95.14 | ↑0.12 | 28.71 | - |
| GDP* (Guo et al., 2021) | ✗ | - | 350 | 94.89 | 95.15 | ↑0.26 | 46.22 | - |
| ChipNet* (Tiwari et al., 2021) | ✗ | ✓ | 300 | 93.55 | 92.58 | ↓0.97 | 20.00 | - |
| **LeanFlex-GKP (ours)** | ✓ | ✗ | 300 | 93.87 | 94.30 | ↑**0.43** | 28.74 | 26.98 |

Table 12: Results of VGG11/13/16 Model on CIFAR10 Dataset

| Method | DA | IP | RB | BA | Pruned | △Acc | ↓MACs | ↓Params |
|---|---|---|---|---|---|---|---|---|
| **VGG11 on CIFAR10** | | | | MACs ≈ 153.5M | | Params ≈ 9.3M | | |
| CC (Li et al., 2021) | ✗ | ✗ | 300 | 92.34 | 92.24 | ↓0.10 | 42.32 | 56.77 |
| L1Norm (Li et al., 2017) | ✓ | ✗ | 300 | 92.34 | 91.77 | ↓0.57 | 41.44 | 35.01 |
| **LeanFlex-GKP (ours)** | ✓ | ✗ | 300 | 92.34 | **92.55** | ↑**0.21** | 43.41 | 43.68 |
| **VGG13 on CIFAR10** | | | | MACs ≈ 229.4M | | Params ≈ 9.4M | | |
| CC (Li et al., 2021) | ✗ | ✗ | 300 | 93.95 | 93.97 | ↑0.02 | 42.56 | 54.11 |
| L1Norm (Li et al., 2017) | ✓ | ✗ | 300 | 93.95 | 93.26 | ↓0.69 | 42.95 | 35.09 |
| **LeanFlex-GKP (ours)** | ✓ | ✗ | 300 | 93.95 | **94.03** | ↑**0.08** | 43.58 | 43.68 |
| **VGG16 on CIFAR10** | | | | MACs ≈ 313.4M | | Params ≈ 14.7M | | |
| CC (Li et al., 2021) | ✗ | ✗ | 300 | 93.94 | 94.14 | ↑0.20 | 43.18 | - |
| GAL (Lin et al., 2019b) | ✗ | ✓ | 300 | 93.94 | 91.29 | ↓2.65 | 35.16 | 47.40 |
| HRank (Lin et al., 2020) | ✗ | ✓ | 300 | 93.94 | 93.57 | ↓0.37 | 32.28 | 40.82 |
| L1Norm (Li et al., 2017) | ✓ | ✗ | 300 | 93.94 | 92.88 | ↓1.06 | 42.71 | 37.85 |
| KPGP (Zhang et al., 2022b) | ✓ | ✗ | 300 | 94.27 | 94.17 | ↑0.13 | 43.15 | 43.59 |
| TMI-GKP (Zhong et al., 2022) | ✓ | ✗ | 300 | 93.94 | 94.07 | ↑0.10 | 25.00 | - |
| **LeanFlex-GKP (ours)** | ✓ | ✗ | 300 | 93.94 | **94.15** | ↑**0.21** | 43.15 | 43.59 |

Table 13: Results of DenseNet40 on CIFAR10 Dataset

| Method | DA | IP | RB | BA | Pruned | △Acc | ↓MACs | ↓Params |
|---|---|---|---|---|---|---|---|---|
| **DenseNet40 on CIFAR10** | | | | MACs ≈ 282.2M | | Params ≈ 1.5M | | |
| GAL* (Lin et al., 2019b) | ✗ | ✓ | - | 94.81 | 94.61 | ↓0.20 | 35.30 | 35.60 |
| Slimming* (Liu et al., 2017) | ✗ | ✓ | - | 94.81 | 94.35 | ↓0.46 | 57.60 | 66.30 |
| HRank* (Lin et al., 2020) | ✗ | ✓ | - | 94.81 | 94.24 | ↓0.57 | 41.00 | 36.50 |
| CC *(pr=0.33)* (Li et al., 2021) | ✗ | ✗ | 300 | 94.81 | 94.75 | ↓0.06 | 32.97 | 51.42 |
| CC *(pr=0.50)* (Li et al., 2021) | ✗ | ✗ | 300 | 94.81 | 94.58 | ↓0.23 | 49.85 | 64.48 |
| CC *(pr=0.67)* (Li et al., 2021) | ✗ | ✗ | 300 | 94.81 | 94.22 | ↓0.59 | 66.55 | 75.88 |
| **LeanFlex-GKP *(pr=0.33)* (ours)** | ✓ | ✗ | 300 | 94.81 | **94.99** | ↑**0.18** | 33.08 | 52.68 |
| **LeanFlex-GKP *(pr=0.50)* (ours)** | ✓ | ✗ | 300 | 94.81 | **94.93** | ↑**0.12** | 49.75 | 64.08 |
| **LeanFlex-GKP *(pr=0.67)* (ours)** | ✓ | ✗ | 300 | 94.81 | **94.72** | ↓**0.09** | 66.42 | 75.55 |
| TMI-GKP *(pr=0.52)* (Zhong et al., 2022) | ✓ | ✗ | 300 | 94.66 | 94.76 | ↑0.10 | 52.49 | 57.22 |
| **LeanFlex-GKP *(pr=0.52)* (ours)** | ✓ | ✗ | 300 | 94.66 | **95.11** | ↑**0.45** | 52.49 | 57.22 |

Table 14: Results of ResNet56/110 on CIFAR100 Dataset

| Method | DA | IP | RB | BA | Pruned | △Acc | ↓MACs | ↓Params |
|---|---|---|---|---|---|---|---|---|
| **ResNet56 on CIFAR100** | | | | MACs ≈ 126.567M | | Params ≈ 0.859M | | |
| TMI-GKP (Zhong et al., 2022) | ✓ | ✗ | 300 | 70.85 | 71.11 | ↑0.26 | 43.22 | 43.19 |
| L1Norm-A (Li et al., 2017) | ✓ | ✗ | 300 | 71.53 | 68.61 | ↓2.92 | 43.05 | 40.86 |
| L1Norm-B (Li et al., 2017) | ✓ | ✗ | 300 | 71.53 | 68.32 | ↓3.21 | 42.16 | 48.20 |
| CC (Li et al., 2021) | ✗ | ✗ | 300 | 71.53 | 71.43 | ↓0.10 | 43.52 | 28.52 |
| SFP (He et al., 2018a) | ✗ | ✓ | 300 | 71.53 | 69.80 | ↓1.73 | 44.29 | 44.82 |
| FPGM (He et al., 2019) | ✗ | ✓ | 300 | 71.53 | 69.48 | ↓2.05 | 43.38 | 43.19 |
| NPPM (Gao et al., 2021) | ✗ | ✗ | 300 | 71.53 | 71.57 | ↑0.04 | 33.54 | 13.04 |
| HRank (Lin et al., 2020) | ✗ | ✓ | 300 | 71.53 | 69.84 | ↓1.69 | 37.39 | 31.32 |
| LCCL* (Dong et al., 2017) | ✗ | - | 300 | 71.33 | 68.37 | ↓2.96 | 39.3 | - |
| TAS* (Dong & Yang, 2019) | ✗ | ✗ | 300 | 73.18 | **72.25** | ↓0.93 | 51.3 | - |
| **LeanFlex-GKP (ours)** | ✓ | ✗ | 300 | 71.53 | **72.11** | ↑**0.58** | 43.22 | 43.18 |
| **ResNet110 on CIFAR100** | | | | MACs ≈ 255.001M | | Params ≈ 1.734M | | |
| TMI-GKP (Zhong et al., 2022) | ✓ | ✗ | 300 | 72.99 | 72.79 | ↓0.20 | 43.31 | 43.37 |
| L1Norm-A (Li et al., 2017) | ✓ | ✗ | 300 | 73.20 | 69.85 | ↓3.35 | 43.74 | 44.41 |
| L1Norm-B (Li et al., 2017) | ✓ | ✗ | 300 | 73.20 | 69.32 | ↓3.88 | 42.22 | 51.96 |
| CC (Li et al., 2021) | ✗ | ✗ | 300 | 73.20 | 73.21 | ↑0.01 | 43.43 | 19.78 |
| NPPM (Gao et al., 2021) | ✗ | ✗ | 300 | 73.20 | 72.38 | ↓0.82 | 42.77 | 18.69 |
| LRF (Joo et al., 2021) | ✗ | ✗ | 300 | 73.20 | 73.58 | ↑0.38 | 43.38 | 42.16 |
| LCCL* (Dong et al., 2017) | ✗ | - | 300 | 72.79 | 70.78 | ↓2.01 | 31.3 | - |
| SFP* (He et al., 2018a) | ✗ | ✓ | 300 | 74.14 | 71.28 | ↓2.86 | 52.3 | - |
| FPGM* (He et al., 2019) | ✗ | ✓ | 300 | 74.14 | 72.55 | ↓1.59 | 52.3 | - |
| TAS* (Dong & Yang, 2019) | ✗ | ✗ | 300 | 75.06 | 73.16 | ↓1.90 | 52.6 | - |
| **LeanFlex-GKP (ours)** | ✓ | ✗ | 300 | 73.20 | **73.63** | ↑**0.43** | 43.31 | 43.36 |

Table 15: Results of ResNet32/56/110 on CIFAR10 dataset with a pruning rate of $\approx 62.5\%$

| Method | DA | IP | RB | BA | Pruned | $\triangle$Acc | $\downarrow$MACs | $\downarrow$Params |
|---|---|---|---|---|---|---|---|---|
| **ResNet32 on CIFAR10** | | | MACs $\approx$ 69.5M | | Params $\approx$ 0.46M | | | |
| L1Norm-A (Li et al., 2017) | ✓ | ✗ | 300 | 92.80 | 89.96 | ↓2.84 | 61.86 | 65.21 |
| L1Norm-B (Li et al., 2017) | ✓ | ✗ | 300 | 92.80 | 90.01 | ↓2.79 | 62.36 | 67.39 |
| CC (Li et al., 2021) | ✗ | ✗ | 300 | 92.80 | 92.39 | ↓0.41 | 61.29 | 54.35 |
| SFP (He et al., 2018a) | ✗ | ✓ | 300 | 92.80 | 90.28 | ↓2.52 | 59.74 | 60.65 |
| FPGM (He et al., 2019) | ✗ | ✓ | 300 | 92.80 | 91.32 | ↓1.48 | 58.28 | 59.57 |
| NPPM (Gao et al., 2021) | ✗ | ✗ | 300 | 92.80 | 91.92 | ↓0.88 | 61.15 | 56.52 |
| DHP (Li et al., 2020) | ✗ | ✓ | 300 | 92.80 | 91.73 | ↓1.07 | 50.92 | - |
| LRF (Joo et al., 2021) | ✗ | ✗ | 300 | 92.80 | 92.79 | ↓0.01 | 56.95 | 56.52 |
| EigenDamage* (Wang et al., 2019) | ✗ | - | - | 95.30 | 95.17 | ↓0.13 | 60 | - |
| **LeanFlex-GKP (ours)** | ✓ | ✗ | 300 | 92.80 | **92.40** | ↓**0.40** | 61.56 | 61.74 |
| **ResNet56 on CIFAR10** | | | MACs $\approx$ 126.6M | | Params $\approx$ 0.85M | | | |
| L1Norm-A (Li et al., 2017) | ✓ | ✗ | 300 | 93.24 | 91.79 | ↓1.45 | 62.43 | 57.64 |
| L1Norm-B (Li et al., 2017) | ✓ | ✗ | 300 | 93.24 | 91.56 | ↓1.68 | 62.25 | 62.35 |
| CC (Li et al., 2021) | ✗ | ✗ | 300 | 93.24 | 93.57 | ↑0.33 | 61.54 | 50.58 |
| SFP (He et al., 2018a) | ✗ | ✓ | 300 | 93.24 | 92.24 | ↓1.00 | 58.61 | 60.24 |
| FPGM (He et al., 2019) | ✗ | ✓ | 300 | 93.24 | 92.64 | ↓0.60 | 58.33 | 59.88 |
| NPPM (Gao et al., 2021) | ✗ | ✗ | 300 | 93.24 | 93.07 | ↓0.17 | 58.49 | 47.05 |
| HRank (Lin et al., 2020) | ✗ | ✓ | 300 | 93.24 | 90.63 | ↓2.61 | 60.56 | 51.88 |
| DHP (Li et al., 2020) | ✗ | ✓ | 300 | 93.24 | 91.66 | ↓1.58 | 60.54 | - |
| AAP* (Zhao et al., 2023) | - | - | - | 92.84 | 92.21 | ↓0.63 | 52.72 | - |
| AMC* (He et al., 2018b) | - | - | - | 92.80 | 91.90 | ↓0.90 | 50.00 | - |
| PFP* (Liebenwein et al., 2020) | ✗ | - | 182 | 92.95 | 93.64 | ↑0.69 | 67.41 | 72.10 |
| **LeanFlex-GKP (ours)** | ✓ | ✗ | 300 | 93.24 | **93.54** | ↑**0.30** | 61.76 | 61.99 |
| **ResNet110 on CIFAR10** | | | MACs $\approx$ 255.0M | | Params $\approx$ 1.73M | | | |
| L1Norm-A (Li et al., 2017) | ✓ | ✗ | 300 | 94.26 | 92.50 | ↓1.76 | 61.58 | 64.16 |
| L1Norm-B (Li et al., 2017) | ✓ | ✗ | 300 | 94.26 | 94.04 | ↓0.22 | 60.29 | 72.25 |
| CC (Li et al., 2021) | ✗ | ✗ | 300 | 94.26 | 94.29 | ↑0.03 | 61.34 | 58.38 |
| SFP (He et al., 2018a) | ✗ | ✓ | 300 | 94.26 | 92.98 | ↓1.28 | 58.70 | 60.29 |
| FPGM (He et al., 2019) | ✗ | ✓ | 300 | 94.26 | 94.11 | ↓0.15 | 58.35 | 60.17 |
| NPPM (Gao et al., 2021) | ✗ | ✗ | 300 | 94.26 | 93.93 | ↓0.33 | 60.81 | 56.87 |
| HRank (Lin et al., 2020) | ✗ | ✓ | 300 | 94.26 | 91.94 | ↓2.32 | 61.90 | 62.49 |
| DHP (Li et al., 2020) | ✗ | ✓ | 300 | 94.26 | 92.73 | ↓1.53 | 74.16 | - |
| LRF (Joo et al., 2021) | ✗ | ✗ | 300 | 94.26 | 94.10 | ↓0.16 | 62.94 | 63.12 |
| ChipNet* (Tiwari et al., 2021) | ✗ | ✓ | 300 | 93.98 | 93.78 | ↓0.20 | 62.41 | - |
| PFP* (Liebenwein et al., 2020) | ✗ | - | 182 | 93.57 | 94.58 | ↑1.01 | 68.94 | 71.98 |
| **LeanFlex-GKP (ours)** | ✓ | ✗ | 300 | 94.26 | **94.35** | ↑**0.09** | 64.22 | 62.19 |

Table 16: Results of ResNet20/32/56/110 on CIFAR10 dataset with a pruning rate of $\approx 43.75\%$. Results in **bold red** indicate being the second best among comparisons.

| Method | DA | IP | RB | BA | Pruned | ΔAcc | ↓ MACs | ↓ Params |
|--------|----|----|----|----|--------|------|--------|----------|
| **ResNet20 on CIFAR10** | | | MACs ≈ 40.9M | | Params ≈ 0.27M | | | |
| TMI-GKP (Zhong et al., 2022) | ✓ | ✗ | 300 | 91.99 | 92.18 | ↑ 0.19 | 42.86 | 43.33 |
| L1Norm-A (Li et al., 2017) | ✓ | ✗ | 300 | 91.99 | 90.54 | ↓ 1.45 | 43.11 | 35.19 |
| L1Norm-B (Li et al., 2017) | ✓ | ✗ | 300 | 91.99 | 90.83 | ↓ 1.16 | 43.87 | 19.63 |
| CC (Li et al., 2021) | ✗ | ✗ | 300 | 91.99 | 91.80 | ↓ 0.19 | 43.47 | 36.30 |
| SFP (He et al., 2018a) | ✗ | ✓ | 300 | 91.99 | 91.15 | ↓ 0.84 | 40.32 | 41.85 |
| FPGM (He et al., 2019) | ✗ | ✓ | 300 | 91.99 | 91.51 | ↓ 0.48 | 43.34 | 43.33 |
| NPPM (Gao et al., 2021) | ✗ | ✗ | 300 | 91.99 | 91.86 | ↓ 0.13 | 43.49 | 35.19 |
| LRF (Joo et al., 2021) | ✗ | ✗ | 300 | 91.99 | 92.23 | ↑ 0.24 | 43.08 | 43.70 |
| DepGraph (Fang et al., 2023) | ✓ | ✗ | 300 | 91.99 | 91.38 | ↓ 0.61 | 42.96 | 41.11 |
| KPGP* (Zhang et al., 2022b) | ✓ | ✗ | 300 | 92.46 | 92.10 | ↓ 0.36 | 55.10 | 55.70 |
| PFP* (Liebenwein et al., 2020) | ✗ | - | 182 | 91.40 | 91.36 | ↓ 0.04 | 32.10 | 43.16 |
| **LeanFlex-GKP (ours)** | ✓ | ✗ | 300 | 91.99 | 92.49 | ↑ 0.50 | 42.86 | 43.33 |
| **ResNet32 on CIFAR10** | | | MACs ≈ 69.5M | | Params ≈ 0.46M | | | |
| TMI-GKP (Zhong et al., 2022) | ✓ | ✗ | 300 | 92.80 | 92.99 | ↑ 0.19 | 43.08 | 43.32 |
| L1Norm-A (Li et al., 2017) | ✓ | ✗ | 300 | 92.80 | 91.45 | ↓ 1.35 | 42.63 | 45.69 |
| L1Norm-B (Li et al., 2017) | ✓ | ✗ | 300 | 92.80 | 91.58 | ↓ 1.22 | 42.96 | 32.54 |
| CC (Li et al., 2021) | ✗ | ✗ | 300 | 92.80 | 93.01 | ↑ 0.21 | 43.49 | 32.76 |
| SFP (He et al., 2018a) | ✗ | ✓ | 300 | 92.80 | 91.94 | ↓ 0.86 | 41.89 | 42.67 |
| FPGM (He et al., 2019) | ✗ | ✓ | 300 | 92.80 | 92.41 | ↓ 0.39 | 43.36 | 43.53 |
| NPPM (Gao et al., 2021) | ✗ | ✗ | 300 | 92.80 | **93.13** | ↑ 0.33 | 43.00 | 29.74 |
| DHP (Li et al., 2020) | ✗ | ✓ | 300 | 92.80 | 92.26 | ↓ 0.54 | 42.30 | 39.01 |
| LRF (Joo et al., 2021) | ✗ | ✗ | 300 | 92.80 | 93.04 | ↑ 0.24 | 44.17 | 43.97 |
| DepGraph (Fang et al., 2023) | ✓ | ✗ | 300 | 92.80 | 93.04 | ↑ 0.24 | 40.79 | 33.26 |
| OTOv2 (Chen et al., 2023) | ✗ | ✓ | 300 | 92.80 | 90.97 | ↓ 1.83 | 38.28 | 44.77 |
| OTOv2 (post-train) (Chen et al., 2023) | ✗ | ✓ | 300 | 92.80 | 92.14 | ↓ 0.66 | 49.77 | 35.80 |
| KPGP* (Zhang et al., 2022b) | ✓ | ✗ | 300 | 92.71 | 92.68 | ↓ 0.03 | 43.1 | 43.4 |
| **LeanFlex-GKP (ours)** | ✓ | ✗ | 300 | 92.80 | 93.01 | ↑ 0.21 | 43.08 | 43.32 |
| **ResNet56 on CIFAR10** | | | MACs ≈ 126.6M | | Params ≈ 0.85M | | | |
| TMI-GKP (Zhong et al., 2022) | ✓ | ✗ | 300 | 93.24 | 93.95 | ↑ 0.71 | 43.23 | 43.49 |
| L1Norm-A (Li et al., 2017) | ✓ | ✗ | 300 | 93.24 | 92.44 | ↓ 0.80 | 46.27 | 42.91 |
| L1Norm-B (Li et al., 2017) | ✓ | ✗ | 300 | 93.24 | 92.62 | ↓ 0.62 | 43.02 | 31.30 |
| CC (Li et al., 2021) | ✗ | ✗ | 300 | 93.24 | **94.04** | ↑ 0.80 | 44.82 | 27.78 |
| SFP (He et al., 2018a) | ✗ | ✓ | 300 | 93.24 | 93.15 | ↓ 0.09 | 43.54 | 43.61 |
| GAL (Lin et al., 2019a) | ✗ | ✓ | 300 | 93.24 | 91.27 | ↓ 1.97 | 22.38 | 17.94 |
| FPGM (He et al., 2019) | ✗ | ✓ | 300 | 93.24 | 93.60 | ↑ 0.36 | 43.38 | 43.49 |
| NPPM (Gao et al., 2021) | ✗ | ✗ | 300 | 93.24 | 93.55 | ↑ 0.21 | 44.02 | 29.54 |
| HRank (Lin et al., 2020) | ✗ | ✓ | 300 | 93.24 | 92.27 | ↓ 0.97 | 37.39 | 31.54 |
| DHP (Li et al., 2020) | ✗ | ✓ | 300 | 93.24 | 92.42 | ↓ 0.82 | 42.09 | 43.73 |
| LRF (Joo et al., 2021) | ✗ | ✗ | 300 | 93.24 | 93.93 | ↑ 0.69 | 43.89 | 42.56 |
| DepGraph (Fang et al., 2023) | ✓ | ✗ | 300 | 93.24 | 93.79 | ↑ 0.55 | 39.82 | 26.71 |
| OTOv2 (Chen et al., 2023) | ✗ | ✓ | 300 | 93.24 | 91.57 | ↓ 1.67 | 36.96 | 43.70 |
| OTOv2 (post-train) (Chen et al., 2023) | ✗ | ✓ | 300 | 93.24 | 93.02 | ↓ 0.22 | 47.70 | 35.01 |
| KPGP* (Zhang et al., 2022b) | ✓ | ✗ | 300 | 93.75 | 93.72 | ↓ 0.03 | 43.2 | 43.5 |
| **LeanFlex-GKP (ours)** | ✓ | ✗ | 300 | 93.24 | **94.00** | ↑ **0.76** | 43.23 | 43.49 |
| **ResNet110 on CIFAR10** | | | MACs ≈ 255.0M | | Params ≈ 1.73M | | | |
| TMI-GKP (Zhong et al., 2022) | ✓ | ✗ | 300 | 94.26 | 94.90 | ↑ 0.64 | 43.31 | 43.52 |
| L1Norm-A (Li et al., 2017) | ✓ | ✗ | 300 | 94.26 | 92.75 | ↓ 1.51 | 43.74 | 44.56 |
| L1Norm-B (Li et al., 2017) | ✓ | ✗ | 300 | 94.26 | 92.96 | ↓ 1.30 | 43.17 | 36.69 |
| CC (Li et al., 2021) | ✗ | ✗ | 300 | 94.26 | 94.31 | ↑ 0.05 | 44.54 | 39.47 |
| SFP (He et al., 2018a) | ✗ | ✓ | 300 | 94.26 | 94.44 | ↑ 0.18 | 43.42 | 43.52 |
| GAL (Lin et al., 2019a) | ✗ | ✓ | 300 | 94.26 | 93.42 | ↓ 0.84 | 29.14 | 31.37 |
| FPGM (He et al., 2019) | ✗ | ✓ | 300 | 94.26 | 94.18 | ↓ 0.08 | 43.39 | 43.52 |
| NPPM (Gao et al., 2021) | ✗ | ✗ | 300 | 94.26 | 94.16 | ↓ 0.10 | 42.46 | 35.19 |
| HRank (Lin et al., 2020) | ✗ | ✓ | 300 | 94.26 | 92.96 | ↓ 1.30 | 18.57 | 5.38 |
| DHP (Li et al., 2020) | ✗ | ✓ | 300 | 94.26 | 92.53 | ↓ 1.73 | 60.25 | 64.58 |
| LRF (Joo et al., 2021) | ✗ | ✗ | 300 | 94.26 | 94.49 | ↑ 0.23 | 43.37 | 42.30 |
| OTOv2 (Chen et al., 2023) | ✗ | ✓ | 300 | 94.26 | 91.58 | ↓ 2.68 | 37.83 | 42.44 |
| OTOv2 (post-train) (Chen et al., 2023) | ✗ | ✓ | 300 | 94.26 | 93.99 | ↓ 0.27 | 38.11 | 42.50 |
| KPGP* (Zhang et al., 2022b) | ✓ | ✗ | 300 | 93.76 | 94.01 | ↑ 0.25 | 43.3 | 43.5 |
| **LeanFlex-GKP (ours)** | ✓ | ✗ | 300 | 94.26 | **94.92** | ↑ **0.66** | 43.31 | 43.52 |

