# OpenReview forum: "LeanFlex-GKP: Advancing Hassle-Free Structured Pruning with Simple Flexible Group Count"
_ICLR.cc/2024/Conference — Submitted to ICLR 2024_

### Official Review · Reviewer_zM1b · 2023-10-29

**Soundness:** 3 good
**Presentation:** 3 good
**Contribution:** 2 fair
**Rating:** 5
**Confidence:** 3

**Summary:**

This paper proposes LeanFlex-GKP, a new structured pruning method that builds on recent work in grouped kernel pruning (GKP). This method is a one-shot, post-train, and data-agnostic. The key idea is to make the number of groups flexible across layers rather than having dynamic operations during filter grouping or pruning stages.

**Strengths:**

- The paper clearly identifies limitations of existing GKP methods in terms of complexity from dynamic operations and proposes a sensible alternative via flexible group counts.
- The method delivers empirical results across a wide range of model architectures and datasets.
- As a one-shot, post-train, data-agnostic technique with minimal hyperparameters, LeanFlex-GKP is far easier to use out-of-the-box compared to many existing methods.

**Weaknesses:**

- The contribution of this paper is limited. This paper seems an incremental work of TMI-GKP, and the main difference between this work and TMI-GKP is the group count evaluation.
- It would be better if the authors can provide the experimental settings, such as hyperparameters they used for different models.
- From my understanding, LeanFlex-GKP also includes dynamic choices of clustering schemes in each of its convolutional layers as TMI-GKP. I would appreciate it if the author can give results of the performance of LeanFlex-GKP and TMI-GKP. I can only get limited information from Table 1 to Table 5.

**Questions:**

- Could the benefits of dynamic group counts extend to other structured pruning granularities like filter or channel pruning?
- Is there an optimal strategy for setting group counts or do they need to be exhaustively evaluated?

---

> ### Author Response · Authors · 2023-11-16
> **Thanks & Initial Response to Reviewer zM1b (1/3)**
>
> We thank the reviewer for finding our method to be sensible, well-motived, and user-friendly, as well as recognizing our effort spent on proving comprehensive empirical evaluations and delivering a clear background introduction. This is almost everything we aimed for, and we are glad to learn our message is well-received. Here, we address your raised concerns and questions.
>
> ### **[W1 - Incremental to TMI-GKP with only different being group count evaluation]: Our method's technical procedure differs from TMI-GKP in every possible stage, and can be applied in scenarios where TMI-GKP can't.**
>
> We respectively disagree with this assessment, though we understand this potential "misunderstanding" might be due to the lack of community recognition of GKP [1] for being a recently proposed pruning granularity awaiting future improvements. We disagree with this assessment as LeanFlex-GKP has a vastly different technical procedure to TMI-GKP; yet, it can be applied in scenarios where TMI-GKP can't (or is not ideal) with many handy characteristics and advantages.
>
> From a technical perspective — as illustrated in [Figure 2](https://openreview.net/pdf?id=YhNXGWVH1N#page=3) of our paper — every Grouped Kernel Pruning (GKP) method has the following stages: *1) Filter grouping, 2) Grouped kernel pruning, and 3) Reconstruction of the pruned model*. **LeanFlex-GKP utilizes a totally different filter grouping and grouped kernel pruning approaches than TMI-GKP.** It considers aspects TMI-GKP does not consider (group count) and proposes an integral optimization to make the connection between stages that TMI-GKP solves individually. **Such designs lead to many performance and efficiency gains**, as demonstrated in [Table 3, 4](https://openreview.net/pdf?id=YhNXGWVH1N#page=17), and [Table 7](https://openreview.net/pdf?id=YhNXGWVH1N#page=18). **Other reviewers also tend to recognize and appreciate the technical novelty of our proposed method** ([R `SmAM`](https://openreview.net/forum?id=YhNXGWVH1N&noteId=uyaRnx6ga1): *"the authors ... exploits GKP with self-designed clustering and pruning methods"*; [R `HKS2`](https://openreview.net/forum?id=YhNXGWVH1N&noteId=ymbBiH0Ih6): The *"idea of making Conv2d flexible under an integral optimization is interesting."*)
>
> The only similarity between the two methods is they both prune grouped kernels, and their pruned model is reconstructed to a grouped convolution format — which are common characteristics shared by almost all proposed methods following the GKP granularity. **We respectfully argue that improving upon a proposed pruning granularity/framework is not incremental in nature.** There are thousands of methods published under the filter pruning granularity/framework [2, 3], bringing impact to the framework and providing massive contributions to the efficiency community. We argue that GKP, being a promising but newly proposed granularity/framework with only a few established methods under its wing, could also use some healthy advancements, as we provided here.
>
> On the practical application aspect, unlike TMI-GKP, LeanFlex-GKP does not require access to model training checkpoints, **making it applicable when such access is restricted (e.g., pruning a pretrained model, which is a pretty common pipeline)**. Also, because we get to "lean down" on the grouping and pruning stages (and especially to avoid the expensive dimensionality reduction operations like k-PCA, as analyzed in [Section 2.1](https://openreview.net/pdf?id=YhNXGWVH1N#page=4)), the efficiency advantage makes LeanFlex-GKP applicable to wide models under a practical context. E.g., with 64-core of EPYC 7742 and an 80G A100 allocated, it takes more than 50 hours to prune just half of the ResNet101 w.r.t. TMI-GKP. In contrast, we can prune full ResNet101 w.r.t. LeanFlex-GKP in just under 2 hours.
>
> **The reviewer has already recognized the adaptability/usability advantage of our method in [*Strength 3*](https://openreview.net/forum?id=YhNXGWVH1N&noteId=jaYFnjLNfn); we hope the above comparative discussion can put such advantage into the context concerning TMI-GKP.**
>
>
> We emphasize that our work adopts different approaches to almost all stages of GKP to TMI-GKP (though still towards the same goals to leverage the GKP granularity) with significant efficiency, adaptability, and usability improvement by solving a common pain point of exiting GKP implementations (dynamic operations). We consider this a worthy contribution and hope the reviewer would appreciate it too.
>
>
>
>
> ### **[W2 - Add experiment setting (e.g., hyperparameters) clarification]: Sure!**
>
> We have now added such a table in [Appendix C.2](https://openreview.net/pdf?id=YhNXGWVH1N#page=17). Given that this information is mostly used for result replications, we will also open source the experiment setting files and model checkpoints should our manuscript be accepted.

---

> ### Author Response · Authors · 2023-11-16
> **Initial Response to Reviewer zM1b  (2/3)**
>
> ### **[W3.1 - LeanFlex-GKP also includes dynamic choices of clustering schemes in each of its convolutional layers as TMI-GKP]: Not really, its clustering operation is deterministic.**
>
> TMI-GKP applies dynamic clustering schemes, as each of its convolutional layers will go through various combinations of dimensionality reduction and clustering techniques (then one clustering result is chosen according to its *tickets magnitude increase (TMI)* score; see [Section 2.1](https://openreview.net/pdf?id=YhNXGWVH1N#page=4) of our paper or [Section 3.2.1 & 3.2.2](https://openreview.net/pdf?id=LdEhiMG9WLO#page=4) of TMI-GKP [1]). In LeanFlex-GKP, we apply the same *KPP-aware filter grouping procedure* illustrated in [Figure 5](https://openreview.net/pdf?id=YhNXGWVH1N#page=7) for every conv layer, so there is no *"dynamic choice of clustering schemes."*
>
> To provide a more faithful rebuttal, we suspect the reviewer perceives LeanFlex-GKP to be dynamic in terms of filter grouping because we'd consider different `Conv2d(groups)` settings, which obviously influence the clustering/grouping results. While we do consider such settings, `Conv2d(groups)` is just one (and the only method-related) hyperparameter to the same grouping operation, instead of multiple other unique filter grouping approaches, as in TMI-GKP.
>
> This should now be crystal clear with our above [*response to [W2]*](https://openreview.net/forum?id=YhNXGWVH1N&noteId=2EmNfWye86); we thank the reviewer for suggesting this clarification. We have further provided an early definition & example of dynamic operation in [Section 1](https://openreview.net/pdf?id=YhNXGWVH1N), as well as a pseudocode walkthrough of our method in [Appendix B.3](https://openreview.net/pdf?id=YhNXGWVH1N#page=16) to facilitate the digestion of our research motivation and proposed procedure.
>
> ### **[W3.2 - Give more performance comparision between LeanFlex-GKP and TMI-GKP]: We have already compared them under 8 (now 11) different setups.**
>
> There were more comparisons done on the two methods in other tables placed in [Appendix D](https://openreview.net/pdf?id=YhNXGWVH1N#page=19). Here, we provide a gap table for your viewing convenience; **we also added three new results** since our initial submission to make a more comprehensive comparison:
>
>
> > $\Delta$Acc gap between TMI-GKP and LeanFlex-GKP in %, `+` indicates LeanFlex-GKP is better.
> >
> | Model-Dataset            | $\Delta$Acc Gap |  Source |
> | - | :-: | :-: |
> | ResNet50-ImageNet        | +0.11                                      | [Table 9](https://openreview.net/pdf?id=YhNXGWVH1N#page=20) |
> | ResNet56-TinyImageNet    | +3.65                                      | [Table 10](https://openreview.net/pdf?id=YhNXGWVH1N#page=20) |
> | ResNet101-TinyImageNet   | +1.57                                      | Table 10 |
> | ResNet20-CIFAR10         | +0.31                                      |  [Table 16](https://openreview.net/pdf?id=YhNXGWVH1N#page=23) |
> | ResNet32-CIFAR10         | +0.02                                      | Table 16 |
> | ResNet56-CIFAR10         | +0.05                                      | Table 16 |
> | ResNet110-CIFAR10        | +0.02                                      | Table 16 |
> | VGG16-CIFAR10            | +0.11                                      |  [Table 12](https://openreview.net/pdf?id=YhNXGWVH1N#page=21) |
> | ResNet56-CIFAR100 (new)  | +0.32                                      |  [Table 14](https://openreview.net/pdf?id=YhNXGWVH1N#page=21) |
> | ResNet110-CIFAR100 (new)  | +0.63                                      | Table 14 |
> | DenseNet40-CIFAR10 (new) | +0.35                                      | [Table 13](https://openreview.net/pdf?id=YhNXGWVH1N#page=21) |
>
> **It is clear that LeanFlex-GKP has a performance advantage to TMI-GKP across 11 different experiment settings.** We'd say the performance between the two methods is indeed close under some particular setups (e.g., ResNet32/56/110-CIFAR10). But, as we noted in [*Section 5 - Discussion and Conclusion*](https://openreview.net/pdf?id=YhNXGWVH1N#page=8), these BasicBlock CifarResNets experiments are getting saturated, as methods with significant performance gaps on more difficult model-dataset combinations tend to show little difference upon them. Our gap table above also supports this observation with much more meaningful gaps observed under harder settings.
>
> At the risk of being redundant (as the reviewer already recognizes it in [*Strength 3*](https://openreview.net/forum?id=YhNXGWVH1N&noteId=jaYFnjLNfn)), we again emphasize that performance improvement is just one aspect of our contribution. LeanFlex-GKP has significant efficiency, adaptability, and usability advantages over TMI-GKP, as detailed in our [*[W1] response*](https://openreview.net/forum?id=YhNXGWVH1N&noteId=2EmNfWye86) above; yet, our dynamic group count design may inspire all future GKP implementations.

---

> ### Author Response · Authors · 2023-11-16
> **Initial Response to Reviewer zM1b (3/3)**
>
> ### **[Q1 - Can dynamic group counts benefits filter/channel pruning?]: Depending on how you want them to work together.**
>
> Since group count `Conv2D(groups)` is a hyperparameter specific to the grouped convolution format, it is incompatible with the standard filter/channel pruning method where the pruned model is not in a grouped convolution format. However, we can do it the other way around, e.g., we may first prune filters to obtain a smaller layer tensor, then apply GKP (with or without dynamic group counts) to the reduced layer tensor.
>
> While this will require a much different pipeline beyond the scope of our paper, we are confident that dynamic group count will provide benefits, as a filter-pruned model is similar to a smaller (unpruned baseline) model, and our dynamic group count design has showcased improvements across different setups, as demonstrated in [Appendix C](https://openreview.net/pdf?id=YhNXGWVH1N#page=16).
>
> ### **[Q2 - Is there an optimal strategy for setting group counts, or do they need to be exhaustively evaluated?]: It must be post-hoc evaluated, but we don't need to find the absolute optimal.**
>
> Given different group count settings will alter the compute graph of the model, within the train-prune-finetune pipeline, the optimal group count can only be trialed and confirmed with post-hoc evaluations. Considering the data-agnostic setting, we believe the reliance on post-hoc evaluation is always necessary.
>
> However, at the risk of being obvious, we'd note the actual task is not to find the absolute optimal group count setting per different layers (which is costly to search in nature since one cannot find out the best setting in a layer by layer manner even with fine-tuning, as it is a combinatorial problem). We just need to make a reliable guess of which group count setting is better via indicators obtained with cheap operations. Our method does precisely that with lightning-fast data-agnostic group/prune procedures, and it is empirically better at finding the suitable group count than many other potential solutions (see [Appendix C](https://openreview.net/pdf?id=YhNXGWVH1N#page=16)).
>
> ---
> [1] Zhong et al., Revisit Kernel Pruning with Lottery Regulated Grouped Convolutions. ICLR 2022
> [2] Zhou et al., Less is More: Towards Compact CNNs. ECCV 2016
> [3] Li et al., Pruning Filters for Efficient ConvNets. ICLR 2017

---

> ### Author Response · Authors · 2023-11-20
> **An invite to discussion; as well as a digested summary of our rebuttal.**
>
> Right now, with `555` (which are borderline rejections for ICLR), we are hoping to receive some engagements given we posted our rebuttal 4 days ago with the discussion deadline coming close (**Nov 22, [no second stage](https://iclr.cc/Conferences/2024/AuthorGuide)** this year). **We especially wish to receive a response from you**, as all other reviewers' questions are mostly factually rooted and can be objectively addressed. But [your W1](https://openreview.net/forum?id=YhNXGWVH1N&noteId=jaYFnjLNfn) touches on the technical novelty/contribution aspect of our paper, which is a multifaced issue that we believe can only be addressed via a faithful discussion.
>
> We understand that the reviewing pressure is heavy, and we all have personal matters to attend to — **which is possibly especially the case for ICLR, given many of us are authors and reviewers at the same time,** — but please excuse us for urging, as we want to make sure your concerns are adequately addressed. To respect your time, here we provide a contextual summary for your digestion convenience.
>
> ---
>
> We believe it is fair to say that, despite a rate of `5`, **your [feedback](https://openreview.net/forum?id=YhNXGWVH1N&noteId=jaYFnjLNfn) are plenty supportive: as you find our proposed method to be *technically sound*** (*"proposes a sensible alternative..."*), ***well-motivated*** (*"the paper clearly identifies limitations of..."*), ***performant*** (*"delivers empirical results across a wide range of model architectures and datasets"*), **and *user-friendly*** (*"LeanFlex-GKP is far easier to use out-of-the-box..."*) — which are basically everything we hope to strike for; **we appreciate your recognition.**
>
>
>
> Here, we venture to categorize your concerns as the following tri-fold: 1) technical novelty/contribution of LeanFlex-GKP over TMI-GKP. 2) performance comparsion between the two, and 3) clarifications regarding LeanFlex-GKP's grouping procedure, its compatibility with filter/channel pruning, and evaluation procedure for optimal group count.
>
>
> ---
> **We believe we have addressed your *concerns #2 & #3* in a head-on manner** with a [gap table](https://openreview.net/forum?id=YhNXGWVH1N&noteId=u3JRZidmLZ) filled with 11 comparative results (where 3 of them are newly added) and some detailed clarifications (on [grouping](https://openreview.net/forum?id=YhNXGWVH1N&noteId=u3JRZidmLZ) and [the rests](https://openreview.net/forum?id=YhNXGWVH1N&noteId=RT4LWmRCrE)). For *concern #1*, we illustrate the **technical procedure of LeanFlex and TMI-GKP actually differs at every possible stage**. Yet, from a practical standpoint, **LeanFlex-GKP can be utilized in scenarios where TMI can't** (e.g., w/o access to training checkpoint, models with wide layers) and is generally a lot more user-friendly due to several hassle-free designs.
>
> The reviewer is indeed correct that both methods do prune at the same granularity and deliver their pruned models in a grouped convolution format; though, we emphasize that this is the general norm for methods following the GKP granularity/framework. We note — though we believe the reviewer is sure aware — **that advancements under the same framework are common and nonetheless contributive** to the community, which is especially the case given GKP's recency (coined in ICLR 2022) compared to established granularities/frameworks like filter pruning.
>
> On this note, we advocate our observations and investigations on dynamic operations, as well as our proposed integral optimization to make connections on previously separated GKP stages, should provide reference to and inspire future structured pruning works.
>
> We hope the reviewer will appreciate our above analysis. Thanks in advance!
>
> Sincerely,
> Paper7227 Authors

---

### Official Review · Reviewer_HKS2 · 2023-11-03

**Soundness:** 2 fair
**Presentation:** 3 good
**Contribution:** 3 good
**Rating:** 5
**Confidence:** 5

**Summary:**

This paper proposes that the best practice to introduce the dynamic operations to GKP is to make Conv2d flexible under an integral optimization, and proposes a one-shot, post-train, data-agnostic GKP method.

**Strengths:**

1. The writing is clear.
2. The background about Different Structured Pruning Granularities, Grouped Kernel Pruning, and Dynamic Structure Pruning is clearly introduced.
3. The idea of making Conv2d flexible under an integral optimization is interesting.

**Weaknesses:**

1. The evaluation lacks comprehensiveness.

- Baselines are restricted. A lot of related works about iterative structured pruning [1-10] and one-shot pruning [11-15] are not compared.

- Most of the evaluation focuses on the ResNet and VGG architectures. It will be better if more model architectures are evaluated, especially small models like MobileNet, EfficientNet, and ShuffleNet.

- In Section 4, it will be better if some insights can be provided, not only listing numbers.

2. The Ablation Study is missing.

3. The structure can be improved. The Introduction occupies too much space. The first 3.5 pages are all about the Introduction.

4. The discussion about the related works is insufficient, such as iterative structured pruning [1-10], one-shot pruning [11-14], Grouped Kernel Pruning [15], and automatic pruning (with little-to-no hyper-parameter tuning) [2, 3].

[1] EigenDamage: Structured Pruning in the Kronecker-Factored Eigenbasis

[2] Automatic Attention Pruning: Improving and Automating Model Pruning using Attentions

[3] Amc: Automl for model compression and acceleration on mobile devices

[4] Layer-adaptive sparsity for the magnitude-based pruning

[5] Chipnet: Budget-aware pruning with heaviside continuous approximations

[6] Provable filter pruning for efficient neural networks

[7] Accelerate CNNs from Three Dimensions: A Comprehensive Pruning Framework

[8] EagleEye: Fast Sub-net Evaluation for Efficient Neural Network Pruning BT

[9] DMCP: Differentiable Markov Channel Pruning for Neural Networks

[10] GDP: Stabilized Neural Network Pruning via Gates with Differentiable Polarization

[11] Only train once: A one-shot neural network training and pruning framework

[12] Evolutionary multi-objective one-shot filter pruning for designing lightweight convolutional neural network

[13] One-shot layer-wise accuracy approximation for layer pruning

[14] One-shot Network Pruning at Initialization with Discriminative Image Patches

[15] Group-based network pruning via nonlinear relationship between convolution filters

**Questions:**

1. Can authors compare the proposed method with more related pruning works [1-14] (See above)?

2. Can some ablation study be provided?

---

> ### Author Response · Authors · 2023-11-16
> **Thanks & Initial Response to Reviewer HKS2 (1/2)**
>
> We appreciate the reviewer for recognizing the clarity of our delivery and finding our proposed solution to be interesting. Here, we address the concerns and questions you raised.
>
> (Additionally, while it is our understanding that the reviewer provided the extensive list of related work without noticing the comparative evaluation we have done in [Appendix D.2](https://openreview.net/pdf?id=YhNXGWVH1N#page=19), we still want to send our gratitude to the reviewer for the time and effort spent — thank you for taking much care in the reviewing process!)
>
>
> ### **[W1 & Q1 - Evaluation lacks comprehensiveness]: We have already compared with 16 popular structured pruning methods. But sure, we can add 16 more.**
>
>
> We kindly direct the reviewer's attention to [Appendix D.2](https://openreview.net/pdf?id=YhNXGWVH1N#page=19), where we have compared 16 structured pruning methods by the time of initial submission (listed as the first part of [Table 8](https://openreview.net/pdf?id=YhNXGWVH1N#page=19)). To our understanding, this is already a pretty comprehensive coverage that is probably more extensive than most (if not all) pruning arts we compared and suggested by Reviewer `HKS2`.
>
> **As the reviewer is interested, we now add comparisons to 16 *more* methods.** 10 of them are from the reviewer's suggestions, yet 6 additional methods are also included during the expansion exploration (respectively listed as the second and the third portion of [Table 8](https://openreview.net/pdf?id=YhNXGWVH1N#page=19)). We also kindly note that some of the suggested methods by the reviewer are incomparable due to reasons like being under a different paradigm (e.g., LAMP [4], for being unstructured) and not having comparable experiment setups (e.g., EMOFP [12], which is MNIST-focused); yet, methods like 3D [7] and OTOv2 [11] are already compared in our initial submission. **We also added MobileNetV2 results in [Table 10](https://openreview.net/pdf?id=YhNXGWVH1N#page=20)**, as requested by the reviewer.
>
> **Now, we are comparing LeanFlex-GKP against 32 methods and evaluating it under 20 different settings.** To the best of our knowledge — and with many of our experiments running upon an identical baseline under a controlled pipeline — we believe our evaluation is considered the most comprehensive in the structured pruning field.
>
> We guess the reviewer was left with the impression that our evaluation lacks comprehensiveness because of the limited results posted in the main text of our initial submission. We now provide an abbreviated table in the main text ([Table 1](https://openreview.net/pdf?id=YhNXGWVH1N#page=9)), as well as the above-mentioned Method Overview table ([Table 8](https://openreview.net/pdf?id=YhNXGWVH1N#page=19)) to illustrate all compared methods. As the reviewer suggested, we also expanded the discussion and provided more insight in [*Section 5 - Discussion and Conclusion*](https://openreview.net/pdf?id=YhNXGWVH1N#page=8) regarding the results posted in [Section 4.1](https://openreview.net/pdf?id=YhNXGWVH1N#page=8). We hope the reviewer may find the new additions helpful.
>
>
> ### **[W2 & Q2 - Ablation study is missing]: It is in Appendix C. We now expand its coverage with better description.**
>
> We kindly direct the reviewer's attention to [Appendix C](https://openreview.net/pdf?id=YhNXGWVH1N#page=16), where we provided a full-scale ablation study at the time of initial submission. For better readability, we have expanded the ablation study coverage, added more descriptions, and placed an anchor in bold text in [*Section 4 - Experiments*](https://openreview.net/pdf?id=YhNXGWVH1N#page=8) to help navigate among related sections.

---

> ### Author Response · Authors · 2023-11-16
> **Initial Response to Reviewer HKS2 (2/2)**
>
> ### **[W3 - Paper structured can be improved with shorter Intro]: Sure, we changed it up a little!**
>
> We agree that our paper has a (possibly overly) prolonged intro & motivation section ([Section 1 and 2](https://openreview.net/pdf?id=YhNXGWVH1N)). However, given our method is built upon specific observations of the recently proposed GKP granularity [16] with only limited exposure, we feel like we owe it to our readers to provide a clear and unified introduction of the GKP framework and its adaptations without asking them to jump among different papers. Should we remove too much of it, our readers might get confused about GKP and none of our contributions matter. **All reviewers, including you, happen to appreciate the clarity of our background introduction**; so we don't want to drastically modify these sections, if possible.
>
> That being said, we have cut off some repeated material in [Section 1 and 2](https://openreview.net/pdf?id=YhNXGWVH1N), added more connections between our methodology and contribution claims in [Section 3](https://openreview.net/pdf?id=YhNXGWVH1N#page=6) (suggested by [Reviewer `SmAm`](https://openreview.net/forum?id=YhNXGWVH1N&noteId=uyaRnx6ga1)), and added more result discussions in [Section 5](https://openreview.net/pdf?id=YhNXGWVH1N#page=8) (suggested by you).
>
> ### **[W4 - Discussion of related work is insufficient]: Agreed, we have now added them.**
>
> As noted in our response to [*[W1 & Q1]*](https://openreview.net/forum?id=YhNXGWVH1N&noteId=mrbcwq5SxY), we have already provided comprehensive coverage of different pruning methods, but we haven't properly discussed them. We have referenced and added such proper discussion in [Appendix A](https://openreview.net/pdf?id=YhNXGWVH1N#page=14) for all compared methods, as well as some landmark work and important aspects of structured pruning.
>
> We were probably too focused on getting the GKP intro right (and from the feedback, we did) and, therefore, lost the big picture a little. We thank the reviewer for pointing this out and directing us to provide a more zoom-out view of the structured pruning field to our readers.
>
> ---
> (here, we follow the citation index initiated by the reviewer for better consistency)
> [4] Lee et al., Layer-adaptive sparsity for the Magnitude-based Pruning. ICLR 2021
> [7] Wang et al., Accelerate CNNs from Three Dimensions: A Comprehensive Pruning Framework. ICML 2021
> [11] Chen et al., OTOv2: Automatic, Generic, User-Friendly. ICLR 2023 — this is `v2` of the reviewer suggested method [11].
> [12] Wu et al., Evolutionary Multi-Objective One-Shot Filter Pruning for Designing Lightweight Convolutional Neural Network. Sensors 2021
> [16] Zhong et al., Revisit Kernel Pruning with Lottery Regulated Grouped Convolutions. ICLR 2022

---

> ### Author Response · Authors · 2023-11-22
> **Your concerns are rather factually rooted; we have met them with head-on responses — mind confirming if everything is resolved?**
>
> Dear Reviewer `HKS2 `,
>
> If we may, we'd venture to categorize your concerns as the following: *1) More baselines; 2) More model architectures; 3) Need ablation studies; 4) Writing structure improvement; and 5) More related work discussion.*
>
> We believe we have
> * **met your *concern #1, #2, and #3* head-on by pointing to the rich comparison (and ablation studies) we conducted at the time of submission, as well as the newly added results.**
>     * Now, with [32 methods compared](https://openreview.net/pdf?id=YhNXGWVH1N#page=19) and [20 results](https://openreview.net/pdf?id=YhNXGWVH1N#page=18) of LeanFlex-GKP reported, we believe we are among the very top of the evaluation department — if not the best — within the structured pruning field.
>     * We also conducted **new experiments on [MobileNetV2](https://openreview.net/pdf?id=YhNXGWVH1N#page=20)** as the reviewer suggested, and improved **the presentation of (of our now expanded) ablation studies** and [abbreviated experiment results](https://openreview.net/pdf?id=YhNXGWVH1N#page=9) in the main text — two reviewers have missed our results previously; this new update should help.
> * addressed your *concern #4* by expanding [Section 3](https://openreview.net/pdf?id=YhNXGWVH1N#page=6) to make more connections to our claimed contributions and cutting down some repetitive material in Section 1 & 2.
> * **added a [one-page walkthrough](https://openreview.net/pdf?id=YhNXGWVH1N#page=14)** of different aspects of structured pruning in the related work sections (*concern #5*).
>
>
> As mentioned in the title, we believe **your concerns are rather *factually rooted* — as in, there is a minimum-to-zero chance of us disagreeing whether the provided rebuttals have addressed the raised concerns.** We, of course, understand the reviewing pressure [16]. However, please excuse us for urging, as we want to ensure your concerns are adequately addressed. **Would you be so kind as to leave us a quick confirmation and maybe consider improving your rating?**
>
> Thanks in advance, and please do let us know if there's more we can answer.
>
> ---
>
> [16] We believe this year's ICLR schedule is a bit non-optimal, as a real-time system combined with a short discussion stage means the authors are motivated to post their rebuttals asap. Meanwhile, the reviewers — who are likely also authors — are also rushing their rebuttals, rendering low engagement despite everyone working around the clock. We are sorry to urge you in such a situation, but we hope the reviewer would understand, given we have a borderline scoring with most raised concerns that can be (and have been) objectively addressed.

---

### Official Review · Reviewer_SmAM · 2023-11-05

**Soundness:** 2 fair
**Presentation:** 2 fair
**Contribution:** 2 fair
**Rating:** 5
**Confidence:** 3

**Summary:**

In this paper, the authors propose a fine-grained pruning approach, which exploits grouped kernel pruning (GKP) with self-designed clustering and pruning methods, corresponding to high performance and general efficient inference speed. The authors identify that the existing methods correspond to coarse-grained pruning with inferior accuracy performance. In addition, they propose a grouping method to enable exploiting general infrastructure with the pruned model. Then, they propose a L-2 geometric method-based grouped kernel pruning method to perform the pruning operation. Furthermore, they exploit a post-pruning group count evaluation to evaluate the pruned model. They conducted experimental comparison with 5 baseline approaches, which demonstrates the advantages of the proposed approach.

**Strengths:**

1. The authors propose a find-grained pruning method that can have higher accuracy performance.
2. The introduction section is detailed with the presentation of the background explanation.
3. The experimental results seems promising.

**Weaknesses:**

1. The structure of the paper can be improved. Sections 1 and 2 are too detailed that Section 3 corresponds to only a small part, which is not enough to explain the major contribution.
2. 5 baseline approaches are compared while some lossless approaches or other baselines can be added as baseline approaches.
3. It is not clear whether the proposed method can achieve lossless pruning. Some theoretical analysis may be beneficial to the paper.
4. Many grammar errors, e.g., an higher ***, they can be run, with in, is determine by, with lowest etc.
5. Dependable experience may be independable experience.

**Questions:**

1. I wonder if "dependable experience" should be independable experience.
2. I wonder if the proposed approach is lossless. In addition, I wonder if the proposed approach can be applied to other structures.
3. I wonder if the authors can compare the proposed approach with lossless pruning methods.

---

> ### Author Response · Authors · 2023-11-16
> **Thanks & Initial Response to Reviewer SmAM (1/2)**
>
> We appreciate the reviewer for recognizing the evaluation and the performance aspect of our proposed method, as well as our detailed background illustration. Here, we address the concerns and questions you raised.
>
> ### **[W1 - Paper structured can be improved (Sec 1&2 are too long and Sec 3 should be expanded)]: Sure, we changed it up a little.**
>
> We agree that our paper has a (possibly overly) prolonged intro & motivation section ([Section 1 and 2](https://openreview.net/pdf?id=YhNXGWVH1N)). However, given our method is built upon specific observations of the recently proposed GKP granularity [1] with only limited exposure, we feel like we owe it to our readers to provide a clear and unified introduction of the GKP framework and its adaptations without asking them to jump among different papers. Should we remove too much of it, our readers might get confused about GKP and none of our contributions matter. **All reviewers, including you, happen to appreciate the clarity of our background introduction**; so we don't want to drastically modify these sections, if possible.
>
> That said, we agree with the reviewer that *[Section 3 - Proposed Method](https://openreview.net/pdf?id=YhNXGWVH1N#page=6)* should be expanded to better connect our claimed contribution and actual methodology. **We added such connections in [Section 3](https://openreview.net/pdf?id=YhNXGWVH1N#page=6) and cut off some repetitive material in [Section 1 and 2](https://openreview.net/pdf?id=YhNXGWVH1N)**; we thank the reviewer for suggesting this.
>
> ### **[W2.1 - More than 5 baselines should be compared]: We have in fact compared 16 (now 32) pruning methods. We now provide such results with better presentation for improved readability.**
>
>
> We kindly direct the reviewer's attention to [Appendix D.1](https://openreview.net/pdf?id=YhNXGWVH1N#page=19), where we have compared 16 structured pruning methods by the time of submission (first part of [Table 8](https://openreview.net/pdf?id=YhNXGWVH1N#page=19)). **Now**, after including the recommended work by [reviewer `HKS2`](https://openreview.net/forum?id=YhNXGWVH1N&noteId=ymbBiH0Ih6), **we are comparing LeanFlex-GKP against 32 pruning methods** (see [Table 8](https://openreview.net/pdf?id=YhNXGWVH1N#page=19)). To the best of our knowledge — and with many of our experiments conducted with the identical baseline under a controlled pipeline — we believe our evaluation is considered the most comprehensive in the structured pruning field.
>
>
> We guess the reviewer was left with the impression that we only compared with 5 baselines because of the limited results posted in the main text of our initial submission. We now provide an abbreviated table in the main text ([Table 1](https://openreview.net/pdf?id=YhNXGWVH1N#page=9)), as well as a Method Overview table ([Table 8](https://openreview.net/pdf?id=YhNXGWVH1N#page=19)) to illustrate all compared methods. We hope the reviewer may find them helpful.
>
>
> ### **[W2.2 & W3.1 & Q2.1 & Q3 - Is the proposed method "lossless"?]: It depends on your definition of "lossless." Good accuracy retention? — Yes; Exact output? — No.**
>
> Almost all pruning methods will alter the output of the original model (unless under some very special and often trivial setups). Thus, if by "lossless," the reviewer means "exact," like FlashAttention [2], then our method is not "lossless."
>
> This conclusion is rather obvious, given we reported $\Delta$Acc in all our experiments, indicating the difference between the unpruned and prune models. To make the rebuttal more faithful, we suspect the reviewer meant to ask if our method has good accuracy retention (maybe even an improvement after prune). **The answer to this question is model-dataset-setting dependent, but we'd say our method can generally provide good accuracy retention under a reasonable setup**. This is evidenced by 17 out of 20 reported results of LeanFlex-GKP showing improvements after pruning, where no other method can provide a $\uparrow$ $\Delta$Acc under the three exception setups.

---

> ### Author Response · Authors · 2023-11-16
> **Initial Response to Reviewer SmAM (2/2)**
>
> ### **[W3.2 - Lack of theoritcal analysis]: Unfortunately, this is beyond the currently avaliable instruments.**
>
>
> As we are sure the reviewer is well aware, most theoretical analyses done on unpruned NNs require settings that are far distant from the actual models used in pruning tasks (e.g., two-layer ultra-wide MLP [3] vs ResNets). The analysis of pruned NNs is arguably harder. To the best of our knowledge, there is no theoretical analysis for data-agnostic pruning methods following the classic train-prune-finetune pipeline, even for the long-proposed popular filter pruning framework [4, 5] — which was introduced almost a decade ago and has thousands of follow-ups.
>
> Given GKP future complicates the process by alternating the compute graph of the original model (from standard conv to grouped conv), providing a faithful theoretical analysis is, unfortunately, beyond the currently available instruments. **Alternatively**, we have provided a rather comprehensive procedural-based ablation study in [Appendix C.1](https://openreview.net/pdf?id=YhNXGWVH1N#page=16), where we controllably swap in/out components of LeanFlex-GKP with other possible procedural recipes, to facilitate the understanding of method. We hope that would also satisfy the reviewer.
>
>
> ### **[W4 - Typos]: Fixed, and thank you for the close read!**
>
> ### **[W5 & Q1 - Should "dependable experience" be "independable experience"?]: Will change to "predictable" for disambiguation.]**
>
> At the end of [Section 2](https://openreview.net/pdf?id=YhNXGWVH1N#page=6), we note that LeanFlex-GKP's pruned size/compute is directly estimable by multiplying the pruning rate by the readings of the original model, thus *"making the whole pruning procedure a standardized and [dependable] experience."* Here, we use the term "dependable" to indicate our method is reliable and predictable in terms of pruned model size/compute estimation (e.g., in contrast to methods like GAL [6], where one needs to trial-and-error different hyperparameter combinations to reach the desired reduction). So, we indeed meant "dependable."
>
> However, we realize this might cause confusion, as "dependable" can imply hard coupling, which is often not a favorable character. Thus, we will change the term to "predictable" for disambiguation. We thank the reviewer for catching this small but vital detail.
>
> ### **[Q2.2 - Can the proposed approach applied to other structures?]: Again, it depends on your definition of "structure." With filter/channel pruning? — Yes; Outsides CNNs? — No.**
>
> Suppose the reviewer is referring to "structures" as model architectures; given that the convolutional kernel is a concept specific to CNN-like architectures, our method cannot apply to other architectures (e.g., transformers), and neither do most filter/channel pruning methods. However, if the reviewer refers to "structures" as pruning structures, then yes: LeanFlex-GKP can work in conjunction with a filter/channel pruning method — e.g., first remove some filters deemed unimportant, then apply GKP to the pruned model.
>
> ---
> [1] Zhong et al., Revisit Kernel Pruning with Lottery Regulated Grouped Convolutions. ICLR 2022
> [2] Dao et al., FlashAttention: Fast and Memory-Efficient Exact Attention with IO-Awareness. NeurIPS 2022
> [3] Du & Zai et al., Gradient Descent Provably Optimizes Over-parameterized Neural Networks. ICLR 2019
> [4] Zhou et al., Less is More: Towards Compact CNNs. ECCV 2016
> [5] Li et al., Pruning Filters for Efficient ConvNets. ICLR 2017
> [6] Lin et al., Towards Optimal Structured CNN Pruning via Generative Adversarial Learning. CVPR 2019

---

> ### Author Response · Authors · 2023-11-22
> **Your concerns are rather factually rooted; we have met them with head-on responses — mind confirming if everything is resolved?**
>
> Dear Reviewer `SmAM `,
>
> If we may, we'd venture to categorize your concerns as the following: *1) More baselines; 2) Clarification on whether the proposed method is lossless and appliable to other structures 3) Cosmetic/structures suggestions regarding typos fixing, Section 3 expansion, and the use of the term "dependable"; 4) Lack of theoretical analysis.*
>
> **We believe we have:**
> * **met your *concern #1* head-on by pointing to the rich comparison we made at the time of submission, as well as the newly added results.**
>     * Now, with [32 methods compared](https://openreview.net/pdf?id=YhNXGWVH1N#page=19) and [20 results](https://openreview.net/pdf?id=YhNXGWVH1N#page=18) of LeanFlex-GKP reported, we believe we are among the very top of the evaluation department — if not the best — within the structured pruning field.
> * **provided a detailed walkthrough** on different potential interpretations **of your *concern #2***; we believe our clarifications are easy to understand and support our work.
> 	* In short, our method has good accuracy retention but is not exact/equivalent to the unpruned network. It can work in conjunction with filter/channel pruning but not on non-CNNs.
>
> * **made direct adjustments** following your *concern #2* by expanding [Section 3](https://openreview.net/pdf?id=YhNXGWVH1N#page=6) to make more connections to our claimed contributions, fixing typos, and replacing the ambiguous "dependable" term with "predictable."
> * **rationally illustrated** why a faithful theoretical analysis of GKP is beyond the currently available instruments (*concern #4*).
>     * Evidenced by the lack of theoretical support for the long-proposed data-agnostic filter pruning framework.
>
> As mentioned in the title, we believe **your concerns are rather *factually rooted* — as in, there is a minimum-to-zero chance of us disagreeing whether the provided rebuttals have addressed the raised concerns.** We, of course, understand the reviewing pressure [7]. However, please excuse us for urging, as we want to ensure your concerns are adequately addressed. **Would you be so kind as to leave us a quick confirmation and maybe consider improving your rating?**
>
> Thanks in advance, and please do let us know if there's more we can answer.
>
> ---
>
> [7] We believe this year's ICLR schedule is a bit non-optimal, as a real-time system combined with a short discussion stage means the authors are motivated to post their rebuttals asap. Meanwhile, the reviewers — who are likely also authors — are also rushing their rebuttals, rendering low engagement despite everyone working around the clock. We are sorry to urge you in such a situation, but we hope the reviewer would understand, given we have a borderline scoring with most raised concerns that can be (and have been) objectively addressed.

---

### Author Response · Authors · 2023-11-23
**A summary, a reminder, and a note to AC.**

Dear AC,

Given our work is rated with a uniform `555` with no reviewer engagement (despite the fact that we posted all our rebuttals **a week ago on Nov 16**), we want to provide this piece to
* Serve as a digested version of the collected reviewers' feedback, as well as our rebuttal, to provide an easier read and hope for some last-minute engagement
* Provide an advocacy of why our work should be considered for acceptance.


We thank all our reviewers for their valuable time and comments. We are glad that many reviewers found:


* **Our approach is well-motivated, technically sound, and user-friendly.**
    * `HKS2`: *"The idea of making Conv2d flexible under an integral optimization is interesting."*
    * `zM1B`: *"The paper clearly identifies limitations of existing GKP methods ... and proposes a sensible alternative via flexible group counts."*
    * `zM1b`: *"As a one-shot, post-train ... LeanFlex-GKP is far easier to use out-of-the-box compared to many existing methods."*
* **Our proposed method is performant.**
    * `SmAM`: *"... demonstrates the advantages of the proposed approach.""*
    * `SmAM`: *"The authors propose... that can have higher accuracy performance."*
    * `SmAM`: *"The experimental results seems promising."*
    * `zM1b`: *"The method delivers empirical results across a wide range of model architectures and datasets."*)
* **Our writing is clear, especially regarding the background portion.**
    * `SmAM`: *"The introduction section is detailed with the presentation of the background explanation."*
    * `HKS2`: *"The writing is clear."*
    * `HKS2`: *"The background about Different Structured Pruning Granularities ... is clearly introduced."*


Honestly, this is everything we'd like to deliver and we are glad that our message has come across — we appreciate the recognition.

---

Other than some cosmetic issues and some factual clarification (mostly revolving around the capability, characteristic, and compatability of LeanFlex-GKP), our reviewers have raised the following concerns/suggestions.

1. **Lack of enough baselines (`SmAM`,`HKS2`) — reviewers missed materials. New results & better presentation provided.**
    * The two reviewers missed a large portion of our 16-method experiment results supplied in the appendix by the time of initial submission. We have now added 16 more methods as baselines and restructured the presentation to provide more results in the main text.
    * Now, with [32 methods compared](https://openreview.net/pdf?id=YhNXGWVH1N#page=19) and [20 results](https://openreview.net/pdf?id=YhNXGWVH1N#page=18) of LeanFlex-GKP reported, we believe we are among the very top of the evaluation department — if not the best — within the structured pruning field.
2. **Paper structure improvement (`SmAM`, `HKS2`) — addressed**
    * The two reviewers are concerned that the Introduction & Motivation sections (§1 & §2) are too long, where R`SmAM` wants a longer Proposed Method section (§3). Given the clarity of our background introduction is praised by all reviewers, we conservatively cut off some repetitive materials in the Introduction & Motivation sections, and expanded the Proposed Method section to make more connections to our contribution claims.
3. **Lack of theoretical analysis  (`SmAM`) — we believe it is beyond currently available instruments**
    * We illustrated why a faithful theoretical analysis of GKP is beyond the currently available instruments, evidenced by the lack of such analysis for the long-proposed data-agnostic filter pruning framework.
4. **More model architecture (`HKS2`) — added MobileNetV2**
    * We added MobileNetV2 as the reviewer suggested. Now, we have a cover of BasicBlock ResNets, BottleNeck ResNets, VGG, DenseNet, and MobileNetV2, making up a comprehensive coverage of model families.
5. **Lack of ablation studies (`HKS2`) — reviewer missed materials. Coverage & presentation updates are done.**
    * The reviewer missed the ablation studies supplied in [Appendix C](https://openreview.net/pdf?id=YhNXGWVH1N#page=18) at the time of initial submission. We have slightly expanded the ablation study coverage and provided them with better descriptions. We also heavily highlight these studies in the main text for interested readers.
6. **More related work discussion (`HKS2`) — now added 1.5 page related work on structured pruning and GKP in particular ([Appendix A](https://openreview.net/pdf?id=YhNXGWVH1N#page=14))**
7. **Lack of hyperparameter settings (`zM1b`) — now supplied.**
    * We now supply such settings in [Appendix C.2](https://openreview.net/pdf?id=YhNXGWVH1N#page=19). There is only one tunable hyperparameter for our method, which is mostly driven by the layer dimension in practice.

---

> ### Author Response · Authors · 2023-11-23
> **A summary, a reminder, and a note to AC. (cont.)**
>
> 8. **Incremental improvement to TMI-GKP (`zM1b`) — they are vastly different except both being grouped kernel pruning (GKP) methods**
>     * We [illustrate](https://openreview.net/forum?id=YhNXGWVH1N&noteId=2EmNfWye86) the technical procedure of LeanFlex and TMI-GKP actually differs at every possible stage. Yet, from a practical standpoint, LeanFlex-GKP can be utilized in scenarios where TMI can't. The only similarity between the two methods is they both follow the GKP framework; given GKP is a pruning granularity like filter pruning, the existence of TMI-GKP should not discount future GKP studies.
>     * We also compared the performance of LeanFlex and TMI-GKP across [11 different experiment settings](https://openreview.net/forum?id=YhNXGWVH1N&noteId=u3JRZidmLZ). LeanFlex-GKP is leading on all of them with some drastic accuracy gap.
>
> ---
>
> We'd argue **all collected concerns of our work are rather *factually rooted* — as in, there is a minimum-to-zero chance of us and a rational reviewer disagreeing whether the provided rebuttals have addressed the raised concerns or not** (maybe except *concern #8* above, as it is indeed subjective. For this, we provide our detailed rebuttal [here](https://openreview.net/forum?id=YhNXGWVH1N&noteId=2EmNfWye86) for the AC's discretion). **Thus, we hope the AC may step in to either encourage some engagement from our reviewers or conduct a fair final critique/analysis of our work in a standalone manner.**
>
> While we understand that this year's ICLR discussion period is particularly short, and our reviewers might all have other matters to attend to (e.g., working on the rebuttals for their submissions). We feel a bit lost because there is virtually nothing more we can improve on, according to the collected feedback; all reviewers have many nice things to say about our work, yet all except one concerns are objectively addressable. We believe we should see meaningful score improvement should there be a reasonable engagement, and we hope the AC may issue a fair verdict despite all our current (still initial) ratings being negative.
>
> Thank in advance.
>
> Sincerely,
> Paper7227 Authors

---

### Meta-Review · Area_Chair_wp6n · 2023-12-05

**Metareview:**

As the authors have pointed out, and unfortunately, the reviewers were not responsive during the rebuttal period. Since the authors have addressed most of their concerns, they should have receieved better feedback.

To address these concerns, I've discussed it with the SAC and we have decided together to appoint one additional emergency expert reviewer to review the paper, the author's response, and the reviews.

The expert reviewer found most of the reviews to be appropriate, but also, that the rebutal ha indeed addressed most of the concerns. They also mentioned that there is a reasonable enough difference between TMI-GKP and the proposed method.

However, the reviewer still had concerns regarding the experimental settings. The authors have pruned extremely over-parameterized networks on relatively small datasets (resnet100 on cifar10) and one can often get similar performance with smaller dense models. They also found it difficult to compare different methods when they reduce the size of the network using different amounts (see Table-1 ResNet56 on Tiny-ImageNet, DHP vs proposed method).

Including everything, the reviewer found the topic interesting and novel. However, they also thought that the experimental setting can be improved. They have also expected that reviewers would have increased their scores to 6 following the rebuttal and recommended giving the paper a score of 6.

 With all of this information in mind, my recommendation would be to reject the paper from ICLR at this point. I would like to apologize for the authors for the lack of response during the rebuttal process, but also to encourage them to reflect the feedback (as well as their own author response) and submit an improved version of their paper to the next venue.

**Justification For Why Not Higher Score:**

Low scores do not meet the ICLR bar. The paper lies out of my area of expertise and I will have to trust the reviewer's judgement.

**Justification For Why Not Lower Score:**

N/a

---

> ### Public Comment · ~Shaochen_Zhong2 · 2024-04-10
> **Thank you for appointing the emergency expert reviewer. Here we address the two mentioned experiment-related concerns for closure.**
>
> Although the reviewing experience is certainly less than pleasant, we authors are grateful that AC, SAC, and an emergency expert reviewer are able to make efforts to give our work a proper look. We are also pleased that the expert reviewer found our rebuttal ***"addressed most of the concerns,"*** our work to be ***"interesting and novel"***, yet, ***"there is a reasonable enough difference between TMI-GKP and the proposed method"*** — which is the only non-factually-rooted concern the reviewers raised.
>
> Given the comment that ***"they have also expected that reviewers would have increased their scores to 6 following the rebuttal and recommended giving the paper a score of 6."*** Our work essentially is `6666` (borderline accepts), so it is certainly unfortunate to receive rejection after all. However, we do understand the difficulty of making a decision if this paper *"lies outside my (AC's) area of expertise"*, and we appreciate the AC for going beyond and above to seek out an emergency expert reviewer.
>
> ---
> For closure, **here we address the two concerns mentioned by the expert reviewer, which are honestly pretty easy to resolve should we have a chance to be aware of them before the decision deadline.** But we digress, as we already complaint enough about the procedure.
>
> As conveyed, the expert reviewer *"still had concerns regarding the experimental settings."* Where:
>
> >  ***1. The authors have pruned extremely over-parameterized networks on relatively small datasets (resnet100 on cifar10) and one can often get similar performance with smaller dense models.***
>
> We note that we provide two different pruning rates (~62.5% and 43.75% pruned) and three/four ResNets (ResNet20/32/56/110) for CIFAR10 experiments ([Table 15&16](https://openreview.net/pdf?id=YhNXGWVH1N#page=23)), as they are extensively featured in many pruning literature. **So if the reviewer finds one particular setup to be too over-parametrized, there are many different setups to look at.** We think this is viewed as a concern likely because we coincidently featured the ResNet110/CIFAR10 exp in our [Table 1](https://openreview.net/pdf?id=YhNXGWVH1N#page=9) in main text, causing the reviewer to believe we only did such an exp; **but in reality, Table 1 is merely an abbreviation of our full experiments due to space constraints.**
>
> On the note that ResNet110/CIFAR10 is too over-parametrized to the point where *"one can often get similar performance with smaller dense models"* — **we respectfully disagree, as a larger pruned model usually performs much better.** Here is a quick chart:
>
> |Model|Baseline Acc.|Pruned Acc.|Macs Preserved|Params Preserved|
> |-|:-:|:-:|:-:|:-:|
> |ResNet110 (pr=62.5%)|94.26|**94.35**|91.24M|0.65M|
> |ResNet32|92.80|-|69.5M|0.46M|
> |ResNet56|93.24|-|126.6M|0.85M|
>
> **It is clear that a pruned ResNet110 has a better *Pruned Acc.* than unpruned ResNet32/56 models. Note that an unpruned ResNet56 is larger than this pruned ResNet110.**
>
>
> ---
> > ***2. They also found it difficult to compare different methods when they reduce the size of the network using different amounts (see Table-1 ResNet56 on Tiny-ImageNet, DHP vs proposed method).***
>
> Here we copy the referred table for convenience:
>
>
> |Method|Baseline Acc.|Pruned Acc.|$\Delta$Acc.|$\downarrow$Macs|$\downarrow$Params|
> |-|:-:|:-:|:-:|:-:|:-:|
> |DHP*|56.55|55.82|$\downarrow$0.73|55.0|46.00|
> |LeanFlex-GKP (ours)|56.13|55.67|$\downarrow$0.46|37.05|36.76|
>
> **The reviewer is correct that this DHP\* is considered overpruned to our LeanFlex-GKP, and thus an unfair comparison**. Upon close inspection, we notice all overpruned entries in *ResNet56 on Tiny-ImageNet* in Table 1 are directly copied from 3D [1] — noted by the "\*" symbol  —  as that is one of the few other pruning works tested on TinyImageNet outside of TMI-GKP [2].
>
> We believe **the authors of 3D might have used a different (and likely wrong) MACs/FLOPs measuring criteria than ours** (where ours is consistent with popular tools like `thop` and `torch.profiler`), as it is unlikely for DHP [3] — a 2020 from-scratch method delivering some of the weakest results in Table 15&16 (where we replicate and control a fair comparsion pipeline) to be within 0.3%$\Delta$ Acc. to LeanFlex-GKP on a harder task.
>
> Unfortunately, 3D didn't opensource their code, so we can never know for sure. **To clear things up, we then ran DHP under our controlled pipeline and got the following:**
>
> |Method|Baseline Acc.|Pruned Acc.|$\Delta$Acc.|$\downarrow$Macs|
> |-|:-:|:-:|:-:|:-:|
> |DHP |56.13|45.73|$\downarrow$10.40|36.42|
> |LeanFlex-GKP (ours)|56.13|**55.67**|$\downarrow$**0.46**|37.05|36.76|
>
> **This shows that DHP is much worse to LeanFlex-GKP when tested in a controlled manner**; which is consistent with our perception of this inspiring but a bit dated method in the view of today.
>
> ---
> [1] Wang et al., Accelerate CNNs ... ICML 2021.
> [2] Zhong et al., Revisit ... Convolutions. ICLR 2022.
> [3] DHP: Li et al., Differentiable ... HyperNetworks. ECCV 2020.

---

### Decision · Program_Chairs · 2024-01-16

Reject